# STRATEGIC FILTERING FOR CONTENT MODERATION: FREE SPEECH OR FREE OF DISTORTION?

## ABSTRACT

User-generated content (UGC) on social media platforms is vulnerable to incitements and manipulations, necessitating effective regulations. To address these challenges, those platforms often deploy automated content moderators tasked with evaluating the harmfulness of UGC and filtering out content that violates established guidelines. However, such moderation inevitably gives rise to strategic responses from users, who strive to express themselves within the confines of guidelines. Such phenomena call for a careful balance between: 1. ensuring freedom of speech — by minimizing the restriction of expression; and 2. reducing social distortion — measured by the total amount of content manipulation. We tackle the problem of optimizing this balance through the lens of mechanism design, aiming at optimizing the trade-off between minimizing social distortion and maximizing free speech. Although determining the optimal trade-off is NP-hard, we propose practical methods to approximate the optimal solution. Additionally, we provide generalization guarantees determining the amount of finite offline data required to approximate the optimal moderator effectively.

## 1 INTRODUCTION

The Internet has fostered a global ecosystem of social interaction, where social media plays a central role. People use platforms to connect with others, engage with news content, share information, and entertain themselves. Yet, the nature of online content and interactions has increasingly raised concerns among policymakers. Social media can be weaponized to advance extremist causes or disseminate misinformation and fake news. For example, social bots were used to manipulate public discourse during the 2016 U.S. presidential election (Bessi & Ferrara, 2016; Boichak et al., 2018). Another growing concern is cyberbullying, which disproportionately targets public figures and targeted groups (O'Regan & Theil, 2020; West, 2014; MICHAEL, 2024). In response to these issues, several European governments have introduced regulations aimed at curbing fake news and hate speech on social platforms (Cauffman & Goanta, 2021). These incidents highlight the need for robust content moderation mechanisms—policies or algorithmic tools implemented by platforms to limit abuse, promote healthy discourse, and enhance user experience (Roberts, 2018; Barocas & Selbst, 2016). However, while content moderation is crucial for managing platform integrity, it also raises significant concerns regarding censorship and freedom of expression (Shaughnessy et al., 2024; Bollinger & Stone, 2022; Brannon, 2019; Young, 2022). Overly restrictive moderation may inadvertently suppress legitimate content and discourage diversity of expression. In this work, we investigate how to algorithmically balance the competing goals of mitigating harmful content and preserving free speech.

A further complication arises from users' strategic responses to public moderation policies. It is widely observed that users actively engage with *harmful social trends*—not only to gain visibility, but also to evade platform enforcement by subtly modifying their content. For instance, users often manipulate hashtags or obscure key terms to disguise the true nature of their posts (Gerrard, 2018). Inspired by recent work in

strategic machine learning (Hardt et al., 2016), we model the interaction between platform moderators and users as a Stackelberg game, explicitly characterizing users' strategic behavior in response to moderation when engaging with harmful trends. The platform's objective is to design content moderation mechanisms that minimize engagement with such trends while accounting for potential strategic manipulations, and to do so in a way that avoids unnecessary content removal and respects individual rights to free expression.

Our work is related to the growing line of research on strategic ML that studies learning from data provided by strategic agents (Dalvi et al., 2004; Dekel et al., 2008; Brückner & Scheffer, 2011). Hardt et al. (2016) introduced the problem of *strategic classification* as a game between a mechanism designer that deploys a classifier and an agent that best responds to the classifier by modifying their features at a cost. Follow-up work studied different variations of this model, in a PAC-learning setting (Sundaram et al., 2023), online learning (Dong et al., 2018; Chen et al., 2020; Ahmadi et al., 2021), incentivizing agents to take improvement actions rather than gaming actions (Kleinberg & Raghavan, 2020; Haghtalab et al., 2020; Alon et al., 2020; Ahmadi et al., 2022), causal learning (Bechavod et al., 2021; Perdomo et al., 2020), fairness (Hu et al., 2019), etc.

## 1.1 OUR RESULTS AND CONTRIBUTIONS:

**User agency modeling.** We propose a behavioral model that captures the agency of social media users engaging with social trends to gain visibility while staying within moderation boundaries, enabling platform designers to frame the optimal moderation as a mechanism design problem.

**Optimizing the trade-off.** We reduce the mechanism design problem of balancing the goals of curbing harmful trends and protecting free speech to a constrained optimization formulation with offline data. We then propose an effective approximation method to compute the optimal content moderator and validate its effectiveness through experiments on synthetic environments.

**Statistical and computational hardness results.** We provide two key theoretical results: (1) a sample complexity bound (Theorem 1) showing that solving the problem using a polynomial number of offline samples leads to good generalization performance across a broad class of moderation functions (Proposition 3); and (2) a computational hardness result establishing that even for the simple linear function class, finding the optimal content moderator is NP-hard (Theorem 2). This inherent computational complexity motivates our approximation-based approach.

**Conceptual contribution.** Beyond technical results, our work offers a key insight for welfare-driven platform designers: the best practice for mitigating harmful social trends is not to simply suppress associated content, but to define a moderation criterion with a carefully chosen threshold such that a large fraction of content lies near its decision boundary. We formalize this intuition, fully characterize the computational challenges, and propose practical approximation methods to address them.

## 2 PROBLEM SETTING

Let $\mathcal{X} \subset \mathbb{R}^d$ denote the feature space of each user's generated content (UGC). Throughout the paper we assume that $\mathcal{X}$ is convex and compact. Our problem formulation is built upon the interplay between a set of of $n$ users on a social media platform and an automated content moderator $\mathcal{M}$, as shown in the following.

**User representation:** A user indexed by $i$ is represented by a tuple $\boldsymbol{u}_i = (\boldsymbol{x}_i, c_i)$, where $\boldsymbol{x}_i \in \mathcal{X}$ is the feature vector of $\boldsymbol{u}_i$'s generated content representing her original intention of expression and $c_i$ denotes the manipulation cost. We consider the case where the user wants to tweak the original message $\boldsymbol{x}_i$ to $\boldsymbol{z}_i$ to better align with a global ongoing social trend $\boldsymbol{e}$, at a marginal cost $c_i$.

**Content moderator:** The role of a content moderator $\mathcal{M}$ is to regulate published content, ensuring it adheres to platform guidelines. Without loss of generality, $\mathcal{M}$ can be regarded as an indicator function $\mathbb{I}[f(\boldsymbol{x}; \boldsymbol{w}) \leq 0]$, where 0 indicates that the content is flagged as problematic and should be moderated, while 1 indicates it is benign. For simplicity, we define the content moderator as the function $f(\boldsymbol{x}; \boldsymbol{w}) : \mathbb{R}^d \to \mathbb{R}$, parameterized by $\boldsymbol{w}$, and refer to the set $\boldsymbol{x} \in \mathcal{X} : f(\boldsymbol{x}; \boldsymbol{w}) \leq 0$ as the benign region associated with $f$. The output of $f$ can be interpreted as a harmfulness score for each content. In this work, we will focus on moderators $f$ that induce convex benign regions.

**User's strategic response:** With the components outlined above, we can formulate a utility function to capture the potential strategic behavior of a user and predict her response when facing a moderator $f(\cdot; \boldsymbol{w})$, given her profile $\boldsymbol{u} = (\boldsymbol{x}, c)$. [1] The following Eq. (1) characterizes the user's utility when modifying her published content from $\boldsymbol{x}$ to $\boldsymbol{z}$:

$$u(\boldsymbol{z}; (\boldsymbol{x}, c), \boldsymbol{e}, f) = \mathbb{I}[f(\boldsymbol{z}) \leq 0] \cdot \boldsymbol{z}^\top \boldsymbol{e} - c\|\boldsymbol{z} - \boldsymbol{x}\|^2, \tag{1}$$

where the first term reflects the gain by aligning with trend $\boldsymbol{e}$ while not being moderated by $f$, and the second terms measures the manipulation cost. The following proposition 1 characterizes the property of the best response $\boldsymbol{z}^*$ that maximizes Eq. (1).

**Proposition 1.** *Denote the best response of user $\boldsymbol{u} = (\boldsymbol{x}, c)$ against a convex content moderator $f(\boldsymbol{z}; \boldsymbol{w})$ by*

$$\boldsymbol{z}^* = \Delta(\boldsymbol{x}, c; \boldsymbol{e}, f) = \arg\max_{\boldsymbol{z} \in \mathcal{X}} u(\boldsymbol{z}; (\boldsymbol{x}, c), \boldsymbol{e}, f), \tag{2}$$

*and let $\boldsymbol{z}' = \boldsymbol{x} + \frac{\boldsymbol{e}}{2c}$. Then $\boldsymbol{z}^*$ always exists and has the following characterizations:*

*1. if $f(\boldsymbol{z}') \leq 0$, $\boldsymbol{z}^* = \boldsymbol{z}'$.*

*2. if $f(\boldsymbol{z}') > 0$ and $f(\boldsymbol{x}) \leq 0$, $\boldsymbol{z}^* = \mathcal{P}_f(\boldsymbol{z}')$, where $\mathcal{P}_f(\boldsymbol{x})$ denotes the $\ell_2$ projection of $\boldsymbol{x}$ on to the hyperplane $\{\boldsymbol{x} \in \mathbb{R}^d : f(\boldsymbol{x}) = 0\}$.*

*3. if $f(\boldsymbol{z}') > 0$ and $f(\boldsymbol{x}) > 0$, $\boldsymbol{z}^* = \boldsymbol{x}$ or $\mathcal{P}_f(\boldsymbol{z}')$, depending on the location of $\boldsymbol{x}$.*

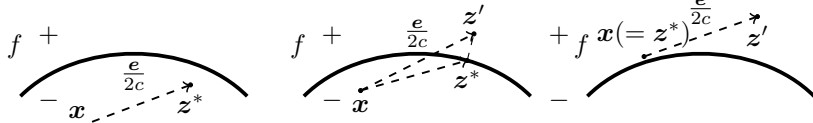

Figure 1: Illustration of best response $\boldsymbol{z}^*$. Left: if $\boldsymbol{z}' = \boldsymbol{x} + \frac{\boldsymbol{e}}{2c}$ is benign, $\boldsymbol{z}^* = \boldsymbol{z}'$. Middle: if $\boldsymbol{x}$ is benign but $\boldsymbol{z}'$ is problematic, $\boldsymbol{x}$ moves to the projection of $\boldsymbol{z}'$ on $f$. Right: if $\boldsymbol{x}$ is already problematic, $\boldsymbol{z}^* = \boldsymbol{x}$, or the projection of $\boldsymbol{z}'$ on $f$, depending on which one yields a higher utility.

Proposition 1 reveals a two-level response pattern. In the first level, each content $\boldsymbol{x}$ tends to shift towards an idealized location $\boldsymbol{z}' = \boldsymbol{x} + \frac{\boldsymbol{e}}{2c}$, manipulating its features in the trending direction $\boldsymbol{e}$ by an amount determined by the cost. Such $\boldsymbol{z}'$ is also the user's preferred manipulation result in the absence of any moderation. The second level can be viewed as a self-correction process starting from $\boldsymbol{z}'$: if $\boldsymbol{z}'$ is accepted by the moderator $f$, it becomes the user's final response; however, if $\boldsymbol{z}'$ is flagged as problematic by $f$, the user would adjust it to the closest point on $f$'s decision boundary, ensuring minimal alteration while still complying with the platform's guidelines. The proof of Proposition 1 is deferred to Appendix B. Proposition 1 highlights the

---

[1]Our utility model connects to previous works, e.g., if we replace the term $\boldsymbol{z}^\top \boldsymbol{e}$ with a user-dependent preference parameter $r$, our model reduces to the agent utility proposed in (Sundaram et al., 2023).

role of content moderation in reducing distortions introduced by the trending direction $\boldsymbol{e}$, which may deviate from users' true expressive intent, represented by $\boldsymbol{x}$. For UGC near the moderation boundary, moderation can mitigate distortion by incentivizing users to align their content with platform guidelines. Thus, the platform can intuitively reduce overall social distortion by encouraging more UGC to move closer to this boundary. However, this strategy comes with a trade-off: the risk of filtering out certain UGC, potentially infringing on users' freedom of expression. This presents a key challenge for the platform—how to balance reducing social distortion with preserving free speech. In the following section, we formalize this problem and explore its complexity and possible solutions.

## 3 THE SOCIAL DISTORTION, FREEDOM OF SPEECH, AND THEIR TRADE-OFF

In this section, we formally introduce the concept of social distortion and explain how content moderation reduces social distortion but at the potential cost of infringing on freedom of speech. From Proposition 1, we observe two major effects of deploying $f$: first, it discourages users from excessively following social trend $\boldsymbol{e}$, which is beneficial; second, it may flag some UGC as harmful, potentially leading to user churn. The first effect can be quantified by the displacement of users who were initially on the benign side. The second effect can be assessed by the proportion of UGC remaining on the platform, serving as an index for freedom of speech. The following Definition 1 formally introduces the concept of social distortion (SD):

**Definition 1.** *The social distortion (SD) of moderator $f$ induced on a user $(\boldsymbol{x}, c)$ is defined as*

$$D(f; (\boldsymbol{x}, c), \boldsymbol{e}) = \begin{cases} \|\boldsymbol{x} - \Delta(\boldsymbol{x}, c; \boldsymbol{e}, f)\|_2^2, & \text{if} \quad f(\boldsymbol{x}) \leq 0, \\ 0, & \text{otherwise.} \end{cases} \tag{3}$$

The social distortion function $D$ defined in Eq. (3) quantifies the manipulation effort of $\boldsymbol{x}$, which measures how much the user's strategic adaptation diverges from her true expressive intent. Importantly, our definition of social distortion applies only to UGC whose original representation $\boldsymbol{x}$ is not filtered by $f$. This is because the strategic behavior of users with $f(\boldsymbol{x}) > 0$ does not contribute to the distortion negatively: as Proposition 1 suggests, users with $f(\boldsymbol{x}) > 0$ are either filtered out—meaning their content is not distorted—or they adjust $\boldsymbol{x}$ to align with platform guidelines, which is considered a beneficial manipulation and thus should not be counted as distortion. Following Definition 1, an immediate observation is that for any user $(\boldsymbol{x}, c)$, deploying $f$ does not increase social distortion relative to an unmoderated environment, as substantiated by the following Proposition 2.

**Proposition 2.** *Let $\perp$ denote a trivial moderator that marks all $\boldsymbol{x} \in \mathcal{X}$ as benign. It holds that*

$$D(f; (\boldsymbol{x}, c), \boldsymbol{e}) \leq D(\perp; (\boldsymbol{x}, c), \boldsymbol{e}), \tag{4}$$

*and the inequality holds strictly if and only if $f(\boldsymbol{x}) \leq 0 < f(\boldsymbol{x} + \frac{\boldsymbol{e}}{2c})$.*

Proposition 2 illustrates a nontrivial moderator can always reduce a user's distortion in her expression, as it requires adjusting $\boldsymbol{x}$ to stay closer to the boundary of $f$ compared to the response $\boldsymbol{x} + \frac{\boldsymbol{e}}{2c}$ induced by a trivial moderator. Based on this, we propose a natural optimization objective that evaluates the expected social distortion mitigated by $f$ across a population of users:

**Definition 2.** *The social Distortion Mitigation (DM) induced by a moderator $f$ over a user $\boldsymbol{u} = (\boldsymbol{x}, c)$ is the difference between the average social distortion induced by a trivial moderator $\perp$ and $f$ on $\boldsymbol{u}$, i.e.,*

$$h(f; \boldsymbol{e}, \boldsymbol{x}, c) = D(\perp; (\boldsymbol{x}, c), \boldsymbol{e}) \cdot \mathbb{I}[f(\boldsymbol{x}) \leq 0] - D(f; (\boldsymbol{x}, c), \boldsymbol{e}), \tag{5}$$

*and the total social distortion mitigation induced by $f$ on a population of users $U = \{u = (\boldsymbol{x}_i, c_i)\}_{i=1}^n$ is thus defined as*

$$DM(f; U) = \sum_{\boldsymbol{u} \in U} h(f, u), \tag{6}$$

*where $\Delta(\boldsymbol{x}, c; f, \boldsymbol{e})$ is defined in Eq. (2).*

Given a class of candidate moderator functions $\mathcal{F} = \{f\}$ and a user distribution $\mathcal{U}$, the problem of optimizing expected social distortion over $\mathcal{U}$ can be formulated as finding an $f \in \mathcal{F}$ that maximizes $DM(f;\mathcal{U})$. However, there is no guarantee on how much freedom of speech the optimal moderator $f$ will sacrifice—that is, how many users may need to be filtered out. The trivial moderator $f = \bot$, which does not filter out any users, cannot mitigate any social distortion. This suggests that any moderator aiming to reduce a reasonable amount of social distortion must inevitably sacrifice some degree of freedom of speech.

To illustrate this trade-off, consider the toy model in Figure 2, where $x$ is uniform distributed in a unit ball centered at the origin, and the social trend is $e = (1, 0)$. Clearly, any reasonable moderator $f$ that maximizes $DM$ would have a decision boundary perpendicular to $e$, as this direction maximizes the deterrent effect of $f$ on users' strategic responses. For each moderator $f$ of the form $x = \theta, \theta \in [-1, 1]$, we can plot the induced social distortion mitigation and a freedom of speech preservation index, which is the fraction of content still allowed on the platform, as shown in the right panel of Figure 2. As $f$ moves from the left margin of $\mathcal{X}$ to the right margin, the social distortion mitigation exhibits an inverted U-shape, while the freedom of speech index consistently increases. This illustrates the trade-off between the two measures. The tension arises because the maximum social distortion mitigation is intuitively achieved when $f$ is positioned where the content distribution is most concentrated, whereas freedom of speech preservation pushes the optimal $f$ toward the margins of the distribution, making it difficult to achieve a doubly optimal moderator. As we

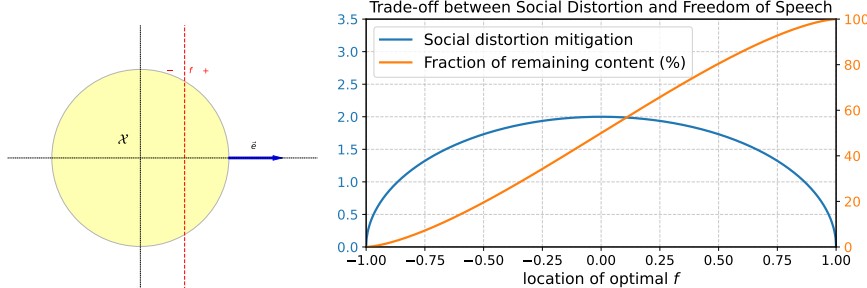

Figure 2: An illustration of the trade-off between social distortion and freedom of speech in a toy model. Left: The original UGC distribution is uniformly random within a unit ball in $\mathbb{R}^2$, with a social trend $e = (1, 0)$. Right: The optimal function $f$ under varying freedom of speech constraints and the resulting induced social distortion.

can learn from the toy example, the key challenge is determining how to strike a balance between the social distortion and freedom of speech objectives, for any possible distribution $\mathcal{U}$. A straightforward approach to is to introduce a hard constraint to the social distortion minimization (or DM maximization) problem, ensuring that at most a certain fraction of users are filtered out had they manipulated their content as much as they wished. More specifically, if a user $x$ would like to follow a social trend $e$ and move to the location $x + \frac{e}{2c}$, but by doing so their content gets filtered out, then their freedom of speech is violated. This leads to the following formalized problem:

$$
\begin{aligned}
\text{find} \quad & \arg\max_{f \in \mathcal{F}} \left\{ \mathbb{E}_{(\boldsymbol{x},c) \sim \mathcal{U}}[h(f;\boldsymbol{x},c)] \right\} \\
\text{subject to} \quad & \mathbb{E}_{(\boldsymbol{x},c) \sim \mathcal{U}}[\mathbb{I}[f(\boldsymbol{x} + \tfrac{\boldsymbol{e}}{2c}) > 0]] \le \theta.
\end{aligned}
\tag{7}
$$

In reality, the platform usually only has access to an offline dataset $U = \{u = (\boldsymbol{x}_i, c_i)\}_{i=1}^n$ sampled from some distribution $\mathcal{U}$. Therefore, a practical way to estimate the solution of OP (7) is to solve the following empirical social distortion optimization problem. During training, we assume that we have access to un-manipulated examples. We can retrieve a set of clean examples by removing part of the content that violates the guidelines, e.g. misinformation.

$$\begin{aligned}
&\text{find} && \arg\max_{f \in \mathcal{F}} \left\{ \sum_{(\boldsymbol{x}_i, c_i) \in S} [h(f; \boldsymbol{x}, c)] \right\} \\
&\text{subject to} && \sum_{i=1}^{n} [\mathbb{I}[f(\boldsymbol{x}_i + \tfrac{\boldsymbol{e}}{2c}) > 0]] \leq K.
\end{aligned} \tag{8}$$

In the following discussion, we will first examine how well the empirical solution to OP (8) approximates the solution to OP (7) using tools from standard statistical learning theory. We will then explore the computational aspects of solving OP (8).

## 4 LEARNABILITY OF GENERAL CONTENT MODERATORS

In this section we establish generalization guarantees for OP (7) – how many samples we need from the true distribution $\mathcal{D}$ to solve OP (8) to approximate the solution of OP (7). Theorem 1 shows sample complexity results in terms of the Vapnik–Chervonenkis dimension (VCDim) of the hypothesis moderator function class $\mathcal{F}$ and *Pseudo-Dimension* (PDim) of its corresponding distortion mitigation function class $\mathcal{H}_{\mathcal{F}}$.

**Theorem 1.** *For any moderator function class $\mathcal{F} = \{f : \mathbb{R}^d \to \{0,1\}\}$ and its induced DM function class $\mathcal{H}_{\mathcal{F}} = \{h(f) | f \in \mathcal{F}\}$ defined by Eq. (5), and any distribution $\mathcal{U}$ on $\mathcal{X} \times \mathcal{C}$, a training sample $U$ of size $\mathcal{O}\left( \frac{1}{\varepsilon^2} \left( H^2(\mathrm{PDim}(\mathcal{H}_{\mathcal{F}}) + \ln(1/\delta)) + \mathrm{VCDim}(\mathcal{F}) \right) \right)$ is sufficient to ensure that with probability at least $1 - \delta$, for every $f \in \mathcal{F}$, the distortion mitigation of $f$ on $U$ and $\mathcal{U}$ and the fraction of filtered points on $U$ and $\mathcal{U}$ each differ by at most $\varepsilon$.*

Intuitively, Pseudo-dimension is a generalization of VC-dimension to real-valued function classes, capturing the capacity of a hypothesis class to fit continuous outputs rather than binary labels. Similar to VC-dimension, which measures the complexity of a class in terms of shattering points in binary classification, pseudo-dimension evaluates the ability of a function class to fit arbitrary real values over a set of points. The formal definition of Pseudo-dimension can be found in Appendix C.

Theorem 1 provides a general yet abstract characterization of the sample complexity required to approximate the solution to OP (7). More concretely, it suggests that problem (7) is statistically learnable if we focus on moderator function classes $\mathcal{F}$ with a finite VC-dimension and ensure that the corresponding class $\mathcal{H}$ has a finite Pseudo-Dimension. Fortunately, many natural function classes $\mathcal{F}$ satisfy these conditions, as shown in the following Proposition 3.

**Proposition 3.** *There exists function classes $\mathcal{F}$ such that $\mathrm{VCDim}(\mathcal{F})$ and $\mathrm{PDim}(\mathcal{H}_{\mathcal{F}})$ are both bounded, such as linear functions, piece-wise linear functions and kernel-based non-linear functions.*

Theorem 1, together with Proposition 3, demonstrates that finding the optimal linear moderator over an offline dataset for Eq. (8) is statistically efficient for many natural and practical function classes, including those discussed in Proposition 3. The linear class is arguably one of the simplest and most effective tools for moderation, capable of representing linear scoring rules that aggregate UGC scores based on relevant features. When combined with feature transformation mappings, linear models can represent techniques like dimensionality reduction followed by linear scoring. Many transformation techniques, such as invertible autoencoders, satisfy the invertibility requirement, ensuring statistical learnability. Additionally, piecewise linear function classes correspond to scenarios where multiple scoring rules are applied simultaneously. However, for such classes, the VCDim and PDim grow exponentially with the number of linear functions. Nevertheless, if the number of functions $m$ remains small, sample-efficient learning is still achievable.

Proof of Theorem 1 (Appendix D) first applies standard learning theory for real-valued functions (Pollard, 1984) to establish a generalization bound for OP (8) without the freedom of speech constraint, relying on the PDim of $\mathcal{H}_{\mathcal{F}}$. Then, a union bound is used to account for the additional constraint, which depends on the VCDim of $\mathcal{F}$. The proof of Proposition 3 involves a detailed analysis of the closed-form best response

mapping for a user facing linear moderators. The core of the proof leverages the Sauer-Shelah Lemma (Sauer, 1972; Shelah, 1972) to establish an upper bound on the PDim for a composition of two function classes. Next, we study the computational complexity of empirically identifying a moderator that optimizes social distortion subject to freedom of speech constraints.

## 5 COMPUTATIONAL TRACTABILITY OF OPTIMAL LINEAR MODERATORS

We discuss the computational complexity of OP (8) in this section. To illustrate the idea, we focus on the class of linear moderator functions (i.e., $\mathcal{F} = \{f(\boldsymbol{x}) = \boldsymbol{w}^\top \boldsymbol{x} + b | \boldsymbol{w} \in \mathbb{R}^d, b \in \mathbb{R}\}$) as it yields a closed-form objective function, which makes the problem more tractable. And in order to also derive a closed-form for the constraint, we use the true feature $\boldsymbol{x}$ to filter content, but not the manipulated feature. Such an easier version of OP (8) is formulated by the following Lemma 1. And perhaps surprisingly, we show that this problem is NP-hard.

**Lemma 1.** *When $\mathcal{F} = \{f(\boldsymbol{x}) = \boldsymbol{w}^\top \boldsymbol{x} + b | \boldsymbol{w} \in \mathbb{R}^d, b \in \mathbb{R}\}$ is the linear function class, OP (8) is equivalent to the following constrained optimization problem:*

$$
\begin{aligned}
&\text{find} && \arg\min\nolimits_{\boldsymbol{w},b} \left\{ \sum_{\{i \in I\}} \left[ (\boldsymbol{w}^\top \boldsymbol{x}_i + b)^2 - \left( \frac{\boldsymbol{w}^\top \boldsymbol{e}}{2c_i} \right)^2 \right] \right\} \\
&\text{subject to} && \sum_{i=1}^n \mathbb{I} \left[ \boldsymbol{w}^\top \left( \boldsymbol{x}_i + \frac{\boldsymbol{e}}{2c_i} \right) + b \leq 0 \right] \geq n - K, \\
& && -1 \leq w_j \leq 1, 1 \leq j \leq d.
\end{aligned}
\tag{9}
$$

*where the index set $I = \{i \in [n] : -\frac{\boldsymbol{w}^\top \boldsymbol{e}}{2c_i} < \boldsymbol{w}^\top \boldsymbol{x}_i + b \leq 0\}$.*

Since OP (9) involves the strict constraint $-\frac{\boldsymbol{w}^\top \boldsymbol{e}}{2c_i} < \boldsymbol{w}^\top \boldsymbol{x}_i + b$, we follow a standard practice by introducing a slack variable $\epsilon > 0$ and consider a relaxed problem, replacing the strict constraint with a non-strict one: $\epsilon - \frac{\boldsymbol{w}^\top \boldsymbol{e}}{2c_i} \leq \boldsymbol{w}^\top \boldsymbol{x}_i + b$. A natural question that follows is whether we can efficiently solve this relaxed version of OP (9). However, despite the nice quadratic form of the objective function in OP (9), the combinatorial nature of the constraint and the indefiniteness of the quadratic objective make the problem challenging to solve. In fact, the next Theorem 2 shows that any $\epsilon$-relaxation of OP (9) is NP-hard.

**Theorem 2.** *For any given input $\epsilon > 0, n, K \in \mathbb{N}_+$ and offline dataset $\mathcal{X} = \{(\boldsymbol{x}_i, c_i)\}_{i=1}^n$, finding the optimal solution to the $\epsilon$-relaxation of OP (9) (by replacing the index set $I$ with $I_\epsilon = \{i \in [n] : \epsilon - \frac{\boldsymbol{w}^\top \boldsymbol{e}}{2c_i} < \boldsymbol{w}^\top \boldsymbol{x}_i + b \leq 0\}$) is NP-hard with respect to $(n, K, 1/\epsilon)$.*

Theorem 2 demonstrates that minimizing social distortion under a hard constraint—limiting the number of users whose content can be filtered—is computationally intractable. This complexity arises because finding a linear moderator $f$ that minimizes social distortion is analogous to finding a hyperplane that maximizes the number of points near its boundary, as the amount of social distortion for each content $\boldsymbol{x}$ only increases as $\boldsymbol{x}$ approaches the boundary of $f$. With the additional constraint, the problem becomes a combinatorial geometric challenge: given a set of $n$ points, find a hyperplane that maximizes the number of points lying on it while ensuring that at least $K$ points remain on each side of the hyperplane. This turns out to be hard. The formal proof, provided in Appendix E.2, contains two core reductions. First, we reduce OP (9) from a combinatorial optimization problem called Maximum Feasible Linear Subsystems (MAX-FLS) with mandatory constraints, and then we show that the problem of MAX-FLS with mandatory constraints is NP-hard by showing a reduction from the Exact 3-Set Cover problem, which is known to be NP-hard.

### 5.1 A HEURISTIC METHOD FOR FINDING APPROXIMATED SOLUTIONS

Since minimizing the social distortion with a hard freedom of speech constraint is NP-hard even for linear function class, we resort to an approximation approach for solving this problem. Still focusing on linear

moderators, a straightforward method for tackling the hard constraint is using penalty functions (Han & Mangasarian, 1979; Nocedal & Wright, 1999). That is, for any $(\boldsymbol{x}_i, c_i)$ that violates the moderator, we introduce a penalty function $P_i(\boldsymbol{w}, b)$ in the objective, as formulated in the following OP:

$$\arg\min_{\boldsymbol{w}, b} \left\{ \sum_{i \in I} \left[ (\boldsymbol{w}^\top \boldsymbol{x}_i + b)^2 - \left( \frac{\boldsymbol{w}^\top \boldsymbol{e}}{2c_i} \right)^2 \right] + \lambda P(\boldsymbol{w}, b) \right\} \tag{10}$$
$$\text{s.t.} \ -1 \leq w_j \leq 1, 1 \leq j \leq d.$$

Let $g(\boldsymbol{w}, b; U, \boldsymbol{e}) = \left[ \boldsymbol{w}^\top \left( \boldsymbol{x}_i + \frac{\boldsymbol{e}}{2c_i} \right) + b > 0 \right]_+$, where the notation $[y_i]_+$ denotes the vector $[\max(0, y_i)]_{i=1}^n$. The hard constraint in OP (9) can thus be expressed as $\|g(\boldsymbol{w}, b; U, \boldsymbol{e})\|_0 \leq K$, where $\|\cdot\|_0$ represents the $\ell_0$-norm. A natural choice for the penalty in OP (10) is then $P(\boldsymbol{w}, b) = \|g(\boldsymbol{w}, b; U, \boldsymbol{e})\|_0$. However, in machine learning and optimization, it is common practice to replace the $\ell_0$ penalty with $\ell_1$- or $\ell_2$-based penalties, often referred to as convex relaxations, to improve numerical stability. We will use the $\ell_2$-based penalty in our experiments.

**Finding the optimal penalty strength $\lambda$:** To balance the social distortion objective and the freedom of speech penalty term, selecting an appropriate parameter $\lambda > 0$ is crucial. A value of $\lambda$ that is too small may fail to yield a solution satisfying the constraint, whereas a sufficiently large $\lambda$ can enforce the constraint but may lead to an excessively large objective value (i.e., a too small DM). Intuitively, there exists an optimal choice of $\lambda$ that strikes this balance. The following proposition demonstrates the existence of such a $\lambda$ and outlines how the platform can determine it.

**Proposition 4.** *Let $(\boldsymbol{w}_1, b_1)$, $(\boldsymbol{w}_2, b_2)$ be any solution of OP (10) by choosing $\lambda = \lambda_1$ and $\lambda = \lambda_2$ ($\lambda_1 > \lambda_2 \geq 0$). Then we have $P(\boldsymbol{w}_1, b_1) \leq P(\boldsymbol{w}_2, b_2)$, $DM(\boldsymbol{w}_1, b_1) \leq DM(\boldsymbol{w}_2, b_2)$.*

Proposition 4 enables the platform to employ a binary search-based approach to determine a proper $\lambda$ for any given constraint $\|g(\boldsymbol{w}, b)\|_0 \leq K$. First, the platform can start with $\lambda_{\max} = \sup_{(\boldsymbol{w}, b)} \{DM(\boldsymbol{w}, b)\} \sim O(n^3)$, as any $\lambda > \lambda_{\max}$ would enforce the equivalent constraint $\|g(\boldsymbol{w}, b)\|_0 = 0$. Then, the platform performs a binary search in interval $[\lambda_{\text{left}}, \lambda_{\text{right}}] = [0, \lambda_{\max}]$. At each iteration, it solves OP (10) with $\lambda = \frac{\lambda_{\text{left}} + \lambda_{\text{right}}}{2}$ and obtains $(\boldsymbol{w}_*, b_*)$. According to Proposition 4, if $P(\boldsymbol{w}_*, b_*) > K$, the platform excludes the lower half of the range $[\lambda_{\text{left}}, \lambda_{\text{right}}]$ in the next round; otherwise, it searches in the upper half. This procedure allows the platform to identify the optimal $\lambda$ within precision $\delta$ by solving at most $O(3 \log_2(n) + \log_2(1/\delta))$ instances of OP (10). The proof of Proposition 4 is straightforward and is provided in Appendix E.3.

**Efficient heuristic with smooth surrogate loss:** Next we discuss for a particular chosen $\lambda$, how to efficiently obtain an solution of OP (10). Let $a_i = \frac{\boldsymbol{w}^\top \boldsymbol{e}}{2c_i}, y_i = \boldsymbol{w}^\top \boldsymbol{x}_i + b + a_i$, we can re-formulate OP (10) as the following cleaner form

$$
\begin{aligned}
&\text{find} && \arg\min_{\boldsymbol{w}, b} \left\{ \sum_{1 \leq i \leq n} l(\boldsymbol{w}, b; \boldsymbol{x}_i, c_i, \boldsymbol{e}) \right\} \\
&\text{subject to} && l(\boldsymbol{w}, b; \boldsymbol{x}_i, c_i, \boldsymbol{e}) = \max\{0, y_i\} \cdot (y_i - 2a_i) + \lambda P_i(\boldsymbol{w}, b) \\
&&& -1 \leq w_j \leq 1, 1 \leq j \leq d.
\end{aligned} \tag{11}
$$

The objective function in OP (11) can be understood as the aggregation of the social good loss $l$ induced by each user $i$, consisting of two components. The first part, $\max\{0, y_i\} \cdot (y_i - 2a_i)$, measures the social distortion incurred by the linear moderator $(\boldsymbol{w}, b)$, and the second part reflects the calibrated infringement on freedom of speech: the larger penalty term $P_i$ is, the farther user $i$'s content is from the decision boundary on the positive side of $f$, making it more likely that user $i$'s content $\boldsymbol{x}$ will be filtered.

The structure of OP (11) resembles the empirical loss minimization problem commonly seen in standard machine learning problems, and we can employ a stochastic gradient descent approach to tackle it, given

any specific penalty functions and trade-off parameter $\lambda$. To ensure the social good loss $l$ is differentiable so that we can apply gradient-based approach, we need to further introduce a surrogate loss $\tilde{l}$ to smooth the non-differentiable point at $y_i = 0$ of $\max\{0, y_i\} \cdot (y_i - 2a_i)$ while selecting a differentiable penalty function. The detail of this treatment is similar to (Levanon & Rosenfeld, 2021) and is outlined in the optimization solver setup in the next section. In the following experiments, we apply this approach to solve (11) using a synthetic dataset and report the approximate optimal linear moderator for different trade-off parameters $\lambda$.

**Experiment Result:** We use a synthetic dataset and perform experiments of our heuristic optimization method. The details of the dataset and the setup of optimizers can be found in Appendix F.1. A 2-dimensional visualization in Figure 3 illustrates the optimal linear moderators for both a small $\lambda$ ($\lambda = 0.1$) and a larger value ($\lambda = 10.0$). Each user's original content feature $x$ is represented by a blue dot, while its strategic response to the moderator is shown in red. The social trend is $e = (1, 0)$. As the figure shows, a larger $\lambda$ shifts the moderator boundary toward the margin of the content distribution, as desired. This results in fewer pieces of content being filtered, while still achieving a reasonable degree of social distortion mitigation. When $\lambda$ is small, the computed optimal moderator minimizes social distortion but at the expense of infringing on more users' freedom of speech. The right panel displays both the social distortion mitigation (i.e., the negative of the optimal objective value of OP (11)) and a freedom of speech preservation index, measured by the fraction of content that remains on the platform under the regulation of the computed moderators with varying $\lambda$. As shown, the freedom of speech index increases as $\lambda$ grows, while social distortion mitigation follows an inverted U-shape. This suggests a trade-off between these two objectives, similar to the one observed in the toy model 2. Our result indicates our proposed optimization technique can effectively approximate a solution, thus allowing the platform to flexibly balance social distortion and freedom of speech despite of the computational challenge.

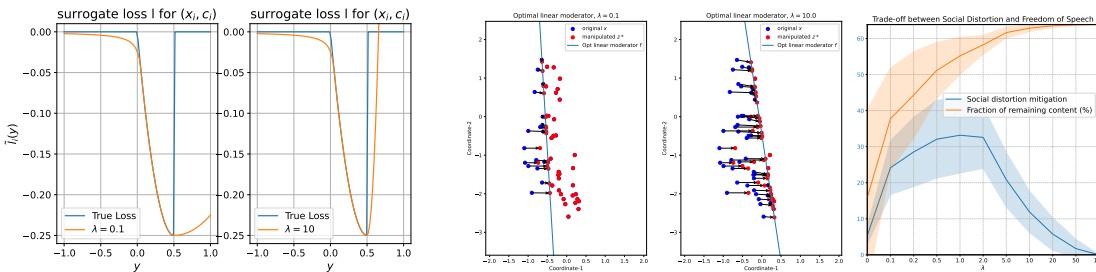

Figure 3: Left: the constructed quasi-convex single point surrogate loss function. Middle: the computed moderator obtained under $\lambda = 0.1$ and $10.0$. Arrows represent users' strategic manipulations against the optimal linear moderator. Right: social distortion mitigation (blue) and the fraction of remaining content on the platform (yellow) incurred by the computed moderator obtained under different $\lambda \in [0.1, 100]$. Error bars obtained from results with 20 independently generated dataset. Error bars are $1\sigma$ region based on results from 20 independently generated datasets.

## 6 CONCLUSION

We addressed the challenge of designing content moderators that reduce engagement with harmful social trends while preserving freedom of speech. By modeling the problem as a constrained optimization, we introduced the concept of social distortion and provided generalization guarantees based on the VC-dimension and Pseudo-dimension of the filter function class. While we established the computational hardness of finding optimal linear filters, we provide an empirically efficient approximation approach that enables the platform to achieve any desirable trade-offs. Our findings highlight the need for efficient algorithms and further exploration of more flexible filtering mechanisms.

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

# Appendix to *Strategic Filtering for Content Moderation: Free Speech or Free of Distortion?*

## A ADDITIONAL RELATED WORK

**Welfare-oriented learning.** The key novelty of our work is a shift from mere predictive performance to welfare-oriented optimization in strategic learning, aligning with a recent call by Rosenfeld & Xu (2025). The goal of standard strategic classification tasks is to preserve accuracy despite agents' manipulations while in our setting accuracy is not well-defined as there is no ground-truth labels for UGC. What we advocate is that the platform should directly optimize a welfare objective that captures the externalities of moderation decisions targeting harmful social trends. Our work also connects with a broader literature that examines the societal impacts of strategic classification. For example, Milli et al. (2019) highlights how counter-strategic defenses increase social burden, Kleinberg & Raghavan (2020) explores how classifiers can steer gaming behavior toward socially beneficial efforts, and Yao et al. (2024b;a) studies how recommender systems can improve user welfare in the presence of strategic content creators. We extend this perspective to a new and timely domain — balancing social distortion and freedom of speech in content moderation.

**Content moderation in social media platforms.** To detect abusive content, platforms use a mix of human moderators and automated algorithms. While early moderation relied heavily on human review (Klonick, 2017), most platforms now employ automated filters to handle overtly inappropriate content (Gillespie, 2018), with humans in the loop to address limitations such as poor generalization to out-of-distribution cases. In this work, we assume the harmful trend is known, e.g., election misinformation, and focus on designing mechanisms that discourage user engagement with such trends while preserving freedom of speech.

## B PROOF OF PROPOSITION 1

*Proof.* Let $\mathcal{D} = \{\boldsymbol{z} : f(\boldsymbol{z}; \boldsymbol{w}) \leq 0\}$. According to the definition of convex moderator, $\mathcal{D}$ is a convex set in $\mathbb{R}^d$.

If $\boldsymbol{x} \in \mathcal{D}$, under the formulation of Eq. (1), each user's best response is the solution to the following convex optimization problem (OP)

$$
\begin{aligned}
\text{find} \quad & \boldsymbol{z}^* = \arg\min_{\boldsymbol{z}} \{-\boldsymbol{z}^\top \boldsymbol{e} + c\|\boldsymbol{z} - \boldsymbol{x}\|_2^2\} \\
\text{subject to} \quad & \boldsymbol{z} \in \mathcal{D}.
\end{aligned}
\tag{12}
$$

Observe that the objective function

$$
-\boldsymbol{z}^\top \boldsymbol{e} + c\|\boldsymbol{z} - \boldsymbol{x}\|_2^2 = c\left\|\boldsymbol{z} - \left(\boldsymbol{x} + \frac{\boldsymbol{e}}{2c}\right)\right\|_2^2 - \frac{\boldsymbol{e}^\top \boldsymbol{e}}{4c},
$$

OP (12) is thus equivalent to

$$
\begin{aligned}
\text{find} \quad & \boldsymbol{z}^* = \arg\min_{\boldsymbol{z}} \left\{\left\|\boldsymbol{z} - \left(\boldsymbol{x} + \frac{\boldsymbol{e}}{2c}\right)\right\|_2^2\right\} \\
\text{subject to} \quad & \boldsymbol{z} \in \mathcal{D}.
\end{aligned}
\tag{13}
$$

Let $\boldsymbol{z}' = \boldsymbol{x} + \frac{\boldsymbol{e}}{2c}$. If $\boldsymbol{z}'$ is feasible, i.e., $f(\boldsymbol{z}') \leq 0$, we have $\boldsymbol{z}^* = \boldsymbol{z}'$. Otherwise, by definition $\boldsymbol{z}^*$ is the $\ell_2$ projection of $\boldsymbol{z}'$ on to the decision boundary of $f$.

If $\boldsymbol{x} \notin \mathcal{D}$, staying at $\boldsymbol{x}$ yield 0 utility for $\boldsymbol{u}$. As a result, $\boldsymbol{z}^* = \mathcal{P}_f(\boldsymbol{z}')$ only when $\boldsymbol{z}'$ yields a negative objective value in OP (12). Otherwise, $\boldsymbol{z}^* = \boldsymbol{x}$.

$\square$

## C  DEFINITION OF PSEUDO-DIMENSION

**Definition 3.** *(Pollard's Pseudo-Dimension) A class $\mathcal{F}$ of real-valued functions P-shatters a set of points $\mathcal{X} = \{x_1, x_2, \cdots, x_n\}$ if there exists a set of thresholds $\gamma_1, \gamma_2, \cdots, \gamma_n$ such that for every subset $T \subseteq \mathcal{X}$, there exists a function $f_T \in \mathcal{F}$ such that $f_T(x_i) \geq \gamma_i$ if and only if $x_i \in T$. In other words, all $2^n$ possible above/below patterns are achievable for targets $\gamma_1, \cdots, \gamma_n$. The pseudo-dimension of $\mathcal{F}$, denoted by $\mathrm{PDim}(\mathcal{F})$, is the size of the largest set of points that it P-shatters.*

## D  OMITTED PROOFS IN SECTION 4

### D.1  PROOF OF PROPOSITION 3

In this section we present the proof of Proposition 3, which upper bounds the PDim of the distortion mitigation (DM) function class $\mathcal{H}_{\mathcal{F}}$ given some example moderator function class $\mathcal{F}$. Specifically, we will show that

1. When $\mathcal{F} = \{f(\boldsymbol{x}) = \mathbb{I}[\boldsymbol{w}^\top \boldsymbol{x} + b \leq 0] | (\boldsymbol{w}, b) \in \mathbb{R}^{d+1}\}$ is the linear class, we have

$$\mathrm{VCDim}(\mathcal{F}) \leq d + 1, \quad \mathrm{PDim}(\mathcal{H}_{\mathcal{F}}) \leq \tilde{\mathcal{O}}(d^2), \tag{14}$$

   where $\tilde{\mathcal{O}}$ is the big $O$ notation omitting the log terms.

2. When $\mathcal{F}$ is a piece-wise linear function class with each instance constitutes $m$ linear functions, i.e., $\mathcal{F} = \{f(\boldsymbol{x}) = \mathbb{I}[\boldsymbol{w}_1^\top \boldsymbol{x} + b_1 \leq 0] \vee \cdots \vee \mathbb{I}[\boldsymbol{w}_m^\top \boldsymbol{x} + b_m \leq 0] | (\boldsymbol{w}_i, b_i) \in \mathbb{R}^{d+1}, 1 \leq i \leq m\}$, we have

$$\mathrm{VCDim}(\mathcal{F}) \leq \tilde{\mathcal{O}}(d \cdot 3^m), \quad \mathrm{PDim}(\mathcal{H}_{\mathcal{F}}) \leq \tilde{\mathcal{O}}(d^{m+1} \cdot 3^{2^m}). \tag{15}$$

3. When $\mathcal{F}$ is the linear class defined on some feature transformation mapping $\phi$, i.e., $\mathcal{F} = \{f(\boldsymbol{x}) = \mathbb{I}[\boldsymbol{w}^\top \phi(\boldsymbol{x}) + b \leq 0] | (\boldsymbol{w}, b) \in \mathbb{R}^{d+1}\}$, as long as $\phi$ is invertible and order-preserving, it also holds that

$$\mathrm{VCDim}(\mathcal{F}) \leq d + 1, \quad \mathrm{PDim}(\mathcal{H}_{\mathcal{F}}) \leq \tilde{\mathcal{O}}(d^2). \tag{16}$$

*Proof.* In this proof we derive Pseudo-dimension upper bounds for the three cases listed.

**Case-1:** When $\mathcal{F} = \{f(\boldsymbol{x}) = \mathbb{I}[\boldsymbol{w}^\top \boldsymbol{x} + b \leq 0] | (\boldsymbol{w}, b) \in \mathbb{R}^{d+1}\}$ is the linear functions class, we can without loss of generality let $\|\boldsymbol{w}\|_2 = 1$ since a simultaneous rescaling of $\boldsymbol{w}, b$ does not change the nature of the moderator function and its induced strategic responses. Next, we derive the DM class $\mathcal{H}_{\mathcal{F}}$ as follows. First of all, plugging in the expression of $f$ into the result of Proposition 1, we obtain a user $(\boldsymbol{x}, c)$'s best response as the following:

1. if $\boldsymbol{w}^\top \cdot \left(\boldsymbol{x} + \frac{\boldsymbol{e}}{2c}\right) + b \leq 0$, $\boldsymbol{z}^* = \boldsymbol{x} + \frac{\boldsymbol{e}}{2c}$.

2. if $\boldsymbol{w}^\top \cdot \left(\boldsymbol{x} + \frac{\boldsymbol{e}}{2c}\right) + b > 0$ and $\boldsymbol{w}^\top \boldsymbol{x} + b \leq 0$, $\boldsymbol{z}^* = P_f(\boldsymbol{x} + \frac{\boldsymbol{e}}{2c})$ which has the following closed-form expression

$$\begin{aligned}
\boldsymbol{z}^* &= \boldsymbol{x} + \frac{\boldsymbol{e}}{2c} - \frac{\boldsymbol{w}^\top(\boldsymbol{x} + \frac{\boldsymbol{e}}{2c})\boldsymbol{w}}{\boldsymbol{w}^\top \boldsymbol{w}} - \frac{b\boldsymbol{w}}{\boldsymbol{w}^\top \boldsymbol{w}} \\
&= \boldsymbol{x} + \frac{\boldsymbol{e}}{2c} - \boldsymbol{w}^\top(\boldsymbol{x} + \frac{\boldsymbol{e}}{2c})\boldsymbol{w} - b\boldsymbol{w}.
\end{aligned} \tag{17}$$

By Definition 1 and 2, we can compute each function $h \in \mathcal{H}_\mathcal{F}$ as

$$h(f, \boldsymbol{e}; \boldsymbol{x}, c) = D(\perp; (\boldsymbol{x}, c), \boldsymbol{e}) - D(f; (\boldsymbol{x}, c), \boldsymbol{e})$$

$$= \mathbb{I}[\boldsymbol{w}^\top \boldsymbol{x} + b \leq 0] \cdot \mathbb{I}\left[\boldsymbol{w}^\top \boldsymbol{x} + b > -\frac{\boldsymbol{w}^\top \boldsymbol{e}}{2c}\right] \cdot \left(\left\|\frac{\boldsymbol{e}}{2c}\right\|_2^2 - \left\|\frac{\boldsymbol{e}}{2c} - \boldsymbol{w}^\top(\boldsymbol{x} + \frac{\boldsymbol{e}}{2c})\boldsymbol{w} - b\boldsymbol{w}\right\|_2^2\right)$$

$$= \mathbb{I}[\boldsymbol{w}^\top \boldsymbol{x} + b \leq 0] \cdot \mathbb{I}\left[\boldsymbol{w}^\top \boldsymbol{x} + b > -\frac{\boldsymbol{w}^\top \boldsymbol{e}}{2c}\right] \cdot \left[-(\boldsymbol{w}^\top \boldsymbol{x} + b)^2 + \left(\frac{\boldsymbol{w}^\top \boldsymbol{e}}{2c}\right)^2\right]. \tag{18}$$

For the ease of notation, let's define $\tilde{\boldsymbol{x}} = (\boldsymbol{x}, \frac{1}{2c}) \in \mathbb{R}^{d+1}$ be the extended feature vector for any user data $(\boldsymbol{x}, c)$. By the definition of Pseudo-dimension, for any function class $\mathcal{F} = \{f(\tilde{\boldsymbol{x}}; \boldsymbol{w}, b)|\boldsymbol{w}, b\}$, the $PDim(\mathcal{F})$ can be reduced to the VC dimension of the epigraph of $\mathcal{F}$, i.e.,

$$PDim(\mathcal{F}) = VCdim(\{h(\tilde{\boldsymbol{x}}, y) = \text{sgn}(f(\tilde{\boldsymbol{x}}) - y)|f \in \mathcal{F}, y \in [-1, 1]\}). \tag{19}$$

Let's define the following three function classes

$$\mathcal{H}_1(d) = \left\{h_1(\boldsymbol{x}, c; \boldsymbol{w}, b) = -(\boldsymbol{w}^\top \boldsymbol{x} + b)^2 + \frac{(\boldsymbol{w}^\top \boldsymbol{e})^2}{4c^2}\middle|(\boldsymbol{x}, c) \in \mathbb{R}^{d+1}, (\boldsymbol{w}, b) \in \mathbb{R}^{d+1}\right\}$$

$$= \left\{h_1(\tilde{\boldsymbol{x}}; \boldsymbol{w}, b) = -(\boldsymbol{w}^\top \tilde{\boldsymbol{x}}_{1:d} + b)^2 + (\boldsymbol{w}^\top \boldsymbol{e})^2 \tilde{x}_{d+1}^2\middle|\tilde{\boldsymbol{x}} \in \mathbb{R}^{d+1}, (\boldsymbol{w}, b) \in \mathbb{R}^{d+1}\right\},$$

$$\mathcal{H}_2(d) = \left\{h_2(\boldsymbol{x}, c; \boldsymbol{w}, b) = \mathbb{I}\left[\boldsymbol{w}^\top\left(\boldsymbol{x} + \frac{\boldsymbol{e}}{2c}\right) + b \geq 0\right]\middle|(\boldsymbol{x}, c) \in \mathbb{R}^{d+1}, (\boldsymbol{w}, b) \in \mathbb{R}^{d+1}\right\}$$

$$= \left\{h_2(\tilde{\boldsymbol{x}}; \boldsymbol{w}, b) = \mathbb{I}\left[\boldsymbol{w}^\top \tilde{\boldsymbol{x}}_{1:d} + (\boldsymbol{w}^\top \boldsymbol{e})\tilde{x}_{d+1} + b \geq 0\right]\middle|\tilde{\boldsymbol{x}} \in \mathbb{R}^{d+1}, (\boldsymbol{w}, b) \in \mathbb{R}^{d+1}\right\},$$

$$\mathcal{H}_3(d) = \left\{h_3(\boldsymbol{x}; \boldsymbol{w}, b) = \mathbb{I}\left[\boldsymbol{w}^\top \boldsymbol{x} + b \leq 0\right]\middle|\boldsymbol{x} \in \mathbb{R}^d, (\boldsymbol{w}, b) \in \mathbb{R}^{d+1}\right\},$$

where $\boldsymbol{x}_{1:d}$ denotes the vector that contains the first $d$-dimension of $\boldsymbol{x}$.

Since $h(\boldsymbol{w}, b; \tilde{\boldsymbol{x}}) = h_1 * h_2 * h_3(\tilde{\boldsymbol{x}}; \boldsymbol{w}, b)$, the Pseudo-dimension of $\mathcal{H}_\mathcal{F}$ can be upper bounded by the following

$$PDim(\mathcal{H}_\mathcal{F}) \leq VCdim\left(\left\{h(\tilde{\boldsymbol{x}}, y) = \text{sgn}\left(\prod_{i=1}^3 h_i(\tilde{\boldsymbol{x}}) - y\right)\middle|h_i \in \mathcal{H}_i, y \in [-1, 1], 1 \leq i \leq 3\right\}\right), \tag{20}$$

where the inequality holds because

$$\left\{\text{sgn}\left(h_1 * h_2 * h_3(\tilde{\boldsymbol{x}}; \boldsymbol{w}, b) - y\right)\middle|(\boldsymbol{w}, b) \in \mathbb{R}^{d+1}\right\} \subseteq \left\{\text{sgn}\left(\prod_{i=1}^3 h_i(\tilde{\boldsymbol{x}}) - y\right)\middle|h_i \in \mathcal{H}_i, 1 \leq i \leq 3\right\}.$$

For any function classes $\mathcal{F}, \mathcal{G}$, define $\mathcal{F} \otimes \mathcal{G} = \{f * g|f \in \mathcal{F}, g \in \mathcal{G}\}$. Then Eq. (20) suggests that in order to upper bound $PDim(\mathcal{H}_\mathcal{F})$, it suffices to upper bound $PDim(\mathcal{H}_1 \otimes \mathcal{H}_2 \otimes \mathcal{H}_3)$. Thanks to Lemma 2 which establishes the $PDim$ of the product of two function classes, this can be done by upper bounding $PDim(\mathcal{H}_1), PDim(\mathcal{H}_2), PDim(\mathcal{H}_3)$ separately. In the following we derive the $PDim$ for each function class $\mathcal{H}_i, i = 1, 2, 3$ and then use Lemma 2 to conclude the proof.

**Deriving** $PDim(\mathcal{H}_3)$**:** First of all, since the Pseudo-Dimension for a binary value function class is exactly the VC dimension of the corresponding real-valued function class inside the indicator function, we immediately obtain

$$PDim(\mathcal{H}_3) \leq d + 1, \tag{21}$$

which is the VC dimension for a $d$ dimensional linear function class.

**Deriving** $PDim(\mathcal{H}_2)$**:** For $\mathcal{H}_2$, it holds that

$$\mathcal{H}_2(d) = \left\{ h_2(\tilde{\boldsymbol{x}}; \boldsymbol{w}, b) = \mathbb{I}\left[\boldsymbol{w}^\top \tilde{\boldsymbol{x}}_{1:d} + (\boldsymbol{w}^\top \boldsymbol{e})\tilde{x}_{d+1} + b \geq 0\right] \Big| \tilde{\boldsymbol{x}} \in \mathbb{R}^{d+1}, (\boldsymbol{w}, b) \in \mathbb{R}^{d+1} \right\}$$

$$\subset \left\{ h_2(\tilde{\boldsymbol{x}}; \boldsymbol{w}, w_{d+1}, b) = \mathbb{I}\left[\boldsymbol{w}^\top \tilde{\boldsymbol{x}}_{1:d} + w_{d+1}\tilde{x}_{d+1} + b \geq 0\right] \Big| \tilde{\boldsymbol{x}} \in \mathbb{R}^{d+1}, (\boldsymbol{w}, w_{d+1}, b) \in \mathbb{R}^{d+2} \right\} \tag{22}$$

$$= \left\{ h_2(\tilde{\boldsymbol{x}}; \tilde{\boldsymbol{w}}, b) = \mathbb{I}\left[\tilde{\boldsymbol{w}}^\top \tilde{\boldsymbol{x}} + b \geq 0\right] \Big| \tilde{\boldsymbol{x}} \in \mathbb{R}^{d+1}, (\tilde{\boldsymbol{w}}, b) \in \mathbb{R}^{d+2} \right\},$$

where the subset relationship (22) holds because we relax the parameter $\boldsymbol{w}^\top \boldsymbol{e}$ correlated with $\boldsymbol{w}$ to an additional independent parameter $w_{d+1} \in \mathbb{R}$. This implies that $\mathcal{H}_2(d)$ is a subclass of indicator functions induced by the $d+1$ dimensional linear class. As a result, the Pseudo-Dimension of $\mathcal{H}_2$ must be upper bounded by $d+2$.

**Deriving** $PDim(\mathcal{H}_1)$**:** To derive $PDim(\mathcal{H}_1)$, we first apply the same trick to relax $\boldsymbol{w}^\top \boldsymbol{e}$ to an independent parameter $w_{d+1}$:

$$\mathcal{H}_1(d) = \left\{ h_1(\tilde{\boldsymbol{x}}; \boldsymbol{w}, b) = (\boldsymbol{w}^\top \tilde{\boldsymbol{x}}_{1:d} + b)^2 - (\boldsymbol{w}^\top \boldsymbol{e})^2 \tilde{x}_{d+1}^2 \Big| \tilde{\boldsymbol{x}} \in \mathbb{R}^{d+1}, (\boldsymbol{w}, b) \in \mathbb{R}^{d+1} \right\}$$

$$\subset \left\{ h_1(\tilde{\boldsymbol{x}}; \tilde{\boldsymbol{w}}, b) = (\boldsymbol{w}^\top \tilde{\boldsymbol{x}}_{1:d} + b)^2 - w_{d+1}^2 \tilde{x}_{d+1}^2 \Big| \tilde{\boldsymbol{x}} \in \mathbb{R}^{d+1}, (\tilde{\boldsymbol{w}}, b) \in \mathbb{R}^{d+2} \right\} \triangleq \tilde{\mathcal{H}}_1(d), \tag{23}$$

where $\tilde{\boldsymbol{w}} = (\boldsymbol{w}, w_{d+1})$. Note that each instance in $\tilde{\mathcal{H}}_1(d)$ can be rewritten as

$$h_1(\tilde{\boldsymbol{x}}; \tilde{\boldsymbol{w}}, b) = \sum_{i=1}^{d+1}\sum_{j=1}^{d+1} \psi_{ij}(\tilde{\boldsymbol{w}}, b)\phi_{ij}(\tilde{\boldsymbol{x}}) + \psi_0(\tilde{\boldsymbol{w}}, b)\phi_0(\tilde{\boldsymbol{x}}), \tag{24}$$

where

$$\psi_{ij} = w_i w_j, \phi_{ij} = x_i x_j, 1 \leq i, j \leq d,$$
$$\psi_{d+1,j} = bw_j, \psi_{i,d+1} = bw_i, \phi_{d+1,j} = x_j, \phi_{i,d+1} = x_i, 1 \leq i, j \leq d,$$
$$\psi_{d+1,d+1} = b^2, \phi_{d+1,d+1} = 1, \psi_0 = -w_{d+1}^2, \phi_0 = \tilde{x}_{d+1}^2.$$

Let $\phi(\tilde{\boldsymbol{x}}) = (\phi_{ij}(\tilde{\boldsymbol{x}}), \phi_0(\tilde{\boldsymbol{x}}))_{1 \leq i \leq d+1, 1 \leq j \leq d+1, i+j < 2d+2} \in \mathbb{R}^{(d+1)^2}$. Consider the linear class $\mathcal{L}_{(d+1)^2} = \{l(\boldsymbol{x}; \boldsymbol{w}, b) = \sum_{i=1}^{(d+1)^2} w_i x_i + b | (\boldsymbol{w}, b) \in \mathbb{R}^{(d+1)^2+1}\}$. Then for any $X_m = (\boldsymbol{x}_1, \cdots, \boldsymbol{x}_m), y \in [-1, 1]$, the label patterns of $(\text{sgn}(h_1(\boldsymbol{x}_i) - y))_{i=1}^m$ that $f_1$ can achieve on $X_m$ can also be achieved by $\mathcal{L}_{(d+1)^2}$ on $(\phi(\boldsymbol{x}_1), \cdots, \phi(\boldsymbol{x}_m))$. Therefore, by definition we have

$$PDim(\mathcal{H}_1) = VCdim(\{h(\boldsymbol{x}, y) = \text{sgn}(h_1(\boldsymbol{x}) - y) | h_1 \in \tilde{\mathcal{H}}_1(d), y \in [-1, 1]\})$$
$$\leq VCdim(\{h(\boldsymbol{x}, y) = \text{sgn}(l(\boldsymbol{x}) - y) | l \in \mathcal{L}_{(d+1)^2}, y \in [-1, 1]\})$$
$$= PDim\left(\mathcal{L}_{(d+1)^2}\right) \leq (d+1)^2 + 1, \tag{25}$$

where inequality (25) holds by Theorem 11.6 (The Pseudo-Dimension of linear class) from Anthony et al. (1999).

Finally, from Lemma 2 we conclude that

$$PDim(\mathcal{H}_1 \otimes \mathcal{H}_2) < 3(1 + \log PDim(\mathcal{H}_1) + \log PDim(\mathcal{H}_2))(PDim(\mathcal{H}_1) + PDim(\mathcal{H}_2))$$
$$< 3(1 + 3\log(d+2))(d+2)(d+3) < 12(d+3)^2 \log(d+2),$$

and therefore

$$\begin{aligned}
PDim(\mathcal{H}_{\mathcal{F}}) &\leq PDim(\mathcal{H}_1 \otimes \mathcal{H}_2 \otimes \mathcal{H}_3) \\
&< 3(1 + \log PDim(\mathcal{H}_1 \otimes \mathcal{H}_2) + \log PDim(\mathcal{H}_3))(PDim(\mathcal{H}_1 \otimes \mathcal{H}_2) + PDim(\mathcal{H}_3)) \\
&= 3(1 + \log(12(d+3)^2 \log(d+2)) + \log(d+1))(12(d+3)^2 \log(d+2) + d + 1) \\
&< 3(1 + 6\log(d+3))(13(d+3)^2 \log(d+3)) \\
&< 273(d+3)^2 \log^2(d+3).
\end{aligned}$$

**Case-2:** When $\mathcal{F}$ is a piece-wise linear function class with each instance constitutes $m$ linear functions, i.e.,

$$\mathcal{F} = \{f(\boldsymbol{x}) = \mathbb{I}[\boldsymbol{w}_1^\top \boldsymbol{x} + b_1 \leq 0] \vee \cdots \vee \mathbb{I}[\boldsymbol{w}_m^\top \boldsymbol{x} + b_m \leq 0 | (\boldsymbol{w}_i, b_i) \in \mathbb{R}^{d+1}, 1 \leq i \leq m\},$$

We first upper bound the VC-dimension of $\mathcal{F}$. If we take $\mathcal{F}, \mathcal{F}'$ to be binary function classes in (32) from Lemma 2, the Pdim of $\mathcal{F}$ coincides with the VCDim of $\mathcal{F}$. Hence, for the composition of $m$ linear functions with each VCDim bounded by $d + 1$, the VCDim of the new function is bounded by

$$\tilde{\mathcal{O}}(3(3(3(d+d)+d)+d)+...) \leq \tilde{\mathcal{O}}(3^m \cdot md) \leq \tilde{\mathcal{O}}(d \cdot 3^m),$$

where $\tilde{\mathcal{O}}$ denotes the big $O$ notation omitting the $\log$ terms.

By Definition 1 and 2, we can compute each function $h \in \mathcal{H}_{\mathcal{F}}$ as

$$\begin{aligned}
&h(f, \boldsymbol{e}; \boldsymbol{x}, c) \\
&= D(\perp; (\boldsymbol{x}, c), \boldsymbol{e}) - D(f; (\boldsymbol{x}, c), \boldsymbol{e}) \\
&= \prod_{i=1}^m \mathbb{I}[\boldsymbol{w}_i^\top \boldsymbol{x}_i \leq 0] \cdot \left( \left\| \frac{\boldsymbol{e}}{2c} \right\|_2^2 - \min\left\{ \min_{1 \leq i \leq m} \left\{ \mathbb{I}\left[ \boldsymbol{w}_i^\top \boldsymbol{x} + b_i > -\frac{\boldsymbol{w}_i^\top \boldsymbol{e}}{2c} \right] \cdot \left\| \mathcal{P}^{(i)}\left( \boldsymbol{x} + \frac{\boldsymbol{e}}{2c} \right) - \boldsymbol{x} \right\|_2^2 \right\}, \right.\right. \\
&\left.\left. \min_{1 \leq i,j \leq m} \left\{ \mathbb{I}\left[ \boldsymbol{w}_i^\top \boldsymbol{x} + b_i > -\frac{\boldsymbol{w}_i^\top \boldsymbol{e}}{2c} \right] \cdot \mathbb{I}\left[ \boldsymbol{w}_j^\top \boldsymbol{x} + b_j > -\frac{\boldsymbol{w}_j^\top \boldsymbol{e}}{2c} \right] \cdot \left\| \mathcal{P}^{(i,j)}\left( \boldsymbol{x} + \frac{\boldsymbol{e}}{2c} \right) - \boldsymbol{x} \right\|_2^2 \right\}, \quad (26) \right.\right. \\
&\left.\left. \min_{1 \leq i,j,k \leq m} \left\{ \prod_{t \in \{i,j,k\}} \mathbb{I}\left[ \boldsymbol{w}_t^\top \boldsymbol{x} + b_t > -\frac{\boldsymbol{w}_t^\top \boldsymbol{e}}{2c} \right] \cdot \left\| \mathcal{P}^{(i,j,k)}\left( \boldsymbol{x} + \frac{\boldsymbol{e}}{2c} \right) - \boldsymbol{x} \right\|_2^2 \right\}, \cdots \right\} \right), \quad (27)
\end{aligned}$$

where operator $\mathcal{P}^{(i,j)}$ denotes the $L_2$-projection onto the intersection of hyperplanes $l_i : \boldsymbol{w}_i^\top \boldsymbol{x} + b_i \leq 0$ and $l_j : \boldsymbol{w}_j^\top \boldsymbol{x} + b_j \leq 0$, and $\mathcal{P}^{(i,j,k)}$ denotes the $L_2$-projection onto the intersection of hyperplanes $l_i, l_j, l_k$, and so on. This is because there are in total $2^m$ possibilities in terms of the location of $\boldsymbol{x} + \frac{\boldsymbol{e}}{2c}$'s $L_2$ projection to the convex region denoted by $f(\boldsymbol{x}) = 1$, as $\mathcal{P}_f(\boldsymbol{x} + \frac{\boldsymbol{e}}{2c})$ can be on each hyperplane $l_i$, or on the intersections of any two $l_i, l_j$, or on the intersections of any three $l_i, l_j, l_k$, and so on.

Note that each $\mathcal{P}_f^{(r)}$ (i.e., the projection onto the intersection of $r$ hyperplanes) has a closed-form which is a rational function with polynomial at most $r$. As a result, the Pseudo-dimension of the function class containing all functions like $\left\| \mathcal{P}^{(r)}\left( \boldsymbol{x} + \frac{\boldsymbol{e}}{2c} \right) - \boldsymbol{x} \right\|_2^2$ is at most $\mathcal{O}((rd)^r)$, since $rd$ is the number of parameters each function has. Apply Eq. (31) in Lemma 2, we know the Pdim of the class $\mathbb{I}\left[ \boldsymbol{w}_t^\top \boldsymbol{x} + b_t > -\frac{\boldsymbol{w}_t^\top \boldsymbol{e}}{2c} \right] \cdot \left\| \mathcal{P}^{(r)}\left( \boldsymbol{x} + \frac{\boldsymbol{e}}{2c} \right) - \boldsymbol{x} \right\|_2^2$ is at most $\tilde{\mathcal{O}}(3(d + (rd)^r))$. Continue to apply Eq. (32), we can upper bound the $\min$ of at most $C_m^r$ functions with a Pdim of each at most $\tilde{\mathcal{O}}(3(d + (rd)^r))$ as $3^{C_m^r} \cdot C_m^r \cdot \tilde{\mathcal{O}}(3(d + (rd)^r))$, and the Pdim upper bound for the $\min$ of $(m + 1)$ functions with a Pdim of

each at most $3^{C_m^r} \cdot C_m^r \cdot \tilde{\mathcal{O}}(3(d + (rd)^r)), 1 \le r \le m$ is

$$Pdim(\mathcal{H}_{\mathcal{F}}) \le \mathcal{O}(d \cdot 3^m) \cdot \tilde{\mathcal{O}}\left(3^m \sum_{r=0}^{m} 3^{C_m^r} \cdot C_m^r \cdot \tilde{\mathcal{O}}(3(d + (rd)^r))\right)$$

$$\le \tilde{\mathcal{O}}(d \cdot 3^m) \cdot \tilde{\mathcal{O}}(3^{2^m}) \cdot \tilde{\mathcal{O}}((dm)^m) \le \tilde{\mathcal{O}}(d^{m+1} \cdot 3^{2^m}).$$

**Case-3:** When $\mathcal{F} = \{f(\boldsymbol{x}) = \mathbb{I}[\boldsymbol{w}^\top \phi(\boldsymbol{x}) + b \le 0] | (\boldsymbol{w}, b) \in \mathbb{R}^{d+1}\}$ is the linear functions class with some feature transformation mapping $\phi$, the best response of $(\boldsymbol{x}, c)$ is the solution of the following OP

$$\begin{aligned} \text{find} \qquad & \boldsymbol{z}^* = \arg\min_{\boldsymbol{z}} \left\{\left\|\boldsymbol{z} - \left(\boldsymbol{x} + \tfrac{\boldsymbol{e}}{2c}\right)\right\|_2^2\right\} \\ \text{subject to} \quad & \boldsymbol{w}^\top \phi(\boldsymbol{z}) + b \le 0. \end{aligned} \tag{28}$$

Since $\phi$ is invertible, it is equivalent to

$$\begin{aligned} \text{find} \qquad & \boldsymbol{z}^* = \arg\min_{\boldsymbol{z}} \left\{\left\|\phi^{-1}(\boldsymbol{y}) - \phi^{-1}\left(\phi\left(\left(\boldsymbol{x} + \tfrac{\boldsymbol{e}}{2c}\right)\right)\right)\right\|_2^2\right\} \\ \text{subject to} \quad & \boldsymbol{w}^\top \boldsymbol{y} + b \le 0. \end{aligned} \tag{29}$$

And also because $\phi$ preserves the order of pair-wise $L_2$ distance of any set of points, the solution of OP (29) is equivalent to the solution of

$$\begin{aligned} \text{find} \qquad & \boldsymbol{z}^* = \arg\min_{\boldsymbol{z}} \left\{\left\|\boldsymbol{y} - \phi\left(\left(\boldsymbol{x} + \tfrac{\boldsymbol{e}}{2c}\right)\right)\right\|_2^2\right\} \\ \text{subject to} \quad & \boldsymbol{w}^\top \boldsymbol{y} + b \le 0. \end{aligned} \tag{30}$$

As a result, we can compute $\boldsymbol{z}^*$ the same way as in Case-1 and the VCDim, PDim upper bounds in Case-1 still applies.

$\square$

## D.2 LEMMAS USED IN THE PROOF OF PROPOSITION 3 AND THEIR PROOFS

**Lemma 2.** *For any class of real valued functions* $\mathcal{F}, \mathcal{F}' \subseteq \{f : \mathbb{R}^d \to [-1, 1]\}$ *and binary valued functions* $\mathcal{G} \subseteq \{g : \mathbb{R}^d \to \{0, 1\}\}$, *define* $\mathcal{F} \otimes \mathcal{G} = \{h(\boldsymbol{x}) = f(\boldsymbol{x}) \times g(\boldsymbol{x}) | f \in \mathcal{F}, g \in \mathcal{G}\}$, *and* $\mathcal{F} \ominus \mathcal{F}' = \{h(\boldsymbol{x}) = \min\{f(\boldsymbol{x}), f'(\boldsymbol{x})\} | f \in \mathcal{F}, f' \in \mathcal{F}'\}$. *Then, it holds that*

$$PDim(\mathcal{F} \otimes \mathcal{G}) < 3(1 + \log d_{\mathcal{F}} d_{\mathcal{G}})(d_{\mathcal{F}} + d_{\mathcal{G}}), \tag{31}$$

$$PDim(\mathcal{F} \ominus \mathcal{F}') < 3(1 + \log d_{\mathcal{F}} d_{\mathcal{F}'})(d_{\mathcal{F}} + d_{\mathcal{F}'}), \tag{32}$$

*where* $d_{\mathcal{F}} = PDim(\mathcal{F}), d_{\mathcal{F}'} = PDim(\mathcal{F}'), d_{\mathcal{G}} = PDim(\mathcal{G})$.

*Proof.* By definition, $PDim(\mathcal{F})$ can be reduced to the VC dimension of the epigraph of $\mathcal{F}$, i.e.,

$$PDim(\mathcal{F}) = VCdim(\{h(x, y) = \text{sgn}(f(x) - y) | f \in \mathcal{F}, y \in [-1, 1]\}). \tag{33}$$

Let $\mathcal{X} = \mathbb{R}^d \times [-1, 1]$, consider an arbitrary set of points $X_m = \{(\boldsymbol{x}_i, y_i) \in \mathcal{X}\}_{i=1}^m$ with cardinality $m$ and any binary hypothesis class $\mathcal{H} \subseteq \{h : \mathcal{X} \to \{0, 1\}\}$. Define the maximum shattering number

$$\Pi(m, \mathcal{H}) = \max_{X_m \in \mathcal{X}^m} \{\text{Card}\{(h(\boldsymbol{x}_1, y_1), \cdots, h(\boldsymbol{x}_m, y_m)) \in \{0, 1\}^m | h \in \mathcal{H}\}\}$$

as the total number of label patterns that $\mathcal{H}$ can possibly achieve on $\mathcal{X}$. Next we upper bound the $\Pi(m, \{f * g | f \in \mathcal{F}, g \in \mathcal{G}\})$. For any fixed $X_m = \{(\boldsymbol{x}_i, y_i) \in \mathcal{X}\}_{i=1}^m \in \mathcal{X}^m$, we claim that the binary variable $\text{sgn}(f(\boldsymbol{x}_i)g(\boldsymbol{x}_i) - y_i)$ is determined by three binary variables $\text{sgn}(f(\boldsymbol{x}_i) - y_i)$ and $g(\boldsymbol{x}_i)$. This is because:

1. when $y_i \geq 0$, $f(\boldsymbol{x}_i)g(\boldsymbol{x}_i) \geq y_i$ holds if and only if $f(\boldsymbol{x}_i) \geq y_i$ and $g(\boldsymbol{x}_i) = 1$.

2. when $y_i < 0$, $f(\boldsymbol{x}_i)g(\boldsymbol{x}_i) \geq y_i$ holds if and only if $f(\boldsymbol{x}_i) \geq y_i$ and $g(\boldsymbol{x}_i) = 1$, or $g(\boldsymbol{x}_i) = 0$.

Therefore, any possible label pattern $(\text{sgn}(f(\boldsymbol{x}_1)g(\boldsymbol{x}_1) - y_1), \cdots, \text{sgn}(f(\boldsymbol{x}_m)g(\boldsymbol{x}_m) - y_m)) \in \{0,1\}^m$ is completely determined by the label patterns $(\text{sgn}(f(\boldsymbol{x}_1) - y_1), \cdots, \text{sgn}(f(\boldsymbol{x}_m) - y_m))$ and $(g(\boldsymbol{x}_1), \cdots, g(\boldsymbol{x}_m))$. As a result, it holds that

$$\text{Card}\{(\text{sgn}(f(\boldsymbol{x}_1)g(\boldsymbol{x}_1) - y_1), \cdots, \text{sgn}(f(\boldsymbol{x}_m)g(\boldsymbol{x}_m) - y_m))|f \in \mathcal{F}, g \in \mathcal{G}\}$$
$$\leq \text{Card}\{(\text{sgn}(f(\boldsymbol{x}_1) - y_1), \cdots, \text{sgn}(f(\boldsymbol{x}_m) - y_m))|f \in \mathcal{F}\} \times \text{Card}\{(g(\boldsymbol{x}_1), \cdots, g(\boldsymbol{x}_m))|g \in \mathcal{G}\},$$

which implies

$$\Pi(m, \mathcal{F} \otimes \mathcal{G}) \leq \Pi(m, \mathcal{F}) \times \Pi(m, \mathcal{G}). \tag{34}$$

Using the same argument, we can similarly show that

$$\Pi(m, \mathcal{F} \ominus \mathcal{F}') \leq \Pi(m, \mathcal{F}) \times \Pi(m, \mathcal{F}'). \tag{35}$$

Therefore, to show Eq. (31) and (32), it suffices to show Eq. (31) starting from Eq. (34). According to Sauer–Shelah Lemma (Sauer, 1972; Shelah, 1972), we have

$$\Pi(m, \mathcal{F}) \leq \sum_{i=0}^{VC(\mathcal{F})} \binom{m}{i} \leq \max\{m+1, m^{d_{\mathcal{F}}}\}, \tag{36}$$

where $VC(\mathcal{F})$ denotes the VC dimension of class $\{\text{sgn}(f(x) - y)|f \in \mathcal{F}\}$, which is also the Pseudo dimension of $\mathcal{F}$ (i.e., $d_{\mathcal{F}}$). And the second inequality of Eq. (36) holds because

1. when $d \geq 3$, we have

$$\left(\frac{d}{m}\right)^d \sum_{i=0}^d \binom{m}{i} \leq \sum_{i=0}^d \left(\frac{d}{m}\right)^i \binom{m}{i} \leq \sum_{i=0}^m \left(\frac{d}{m}\right)^i \binom{m}{i} = \left(1 + \frac{d}{m}\right)^m \leq e^d,$$

   and therefore $\sum_{i=0}^d \binom{m}{i} \leq \left(\frac{em}{d}\right)^d < m^d$.

2. when $d = 2$, we have

$$\sum_{i=0}^d \binom{m}{i} = 1 + m + \frac{m(m-1)}{2} \leq m^2, \forall m \geq 2.$$

3. when $d = 1$, we have $\sum_{i=0}^d \binom{m}{i} = 1 + m$.

From Eq. (34) we know $\mathcal{F} \otimes \mathcal{G}$ has bounded Pseudo dimension. Suppose $PDim(\mathcal{F} \otimes \mathcal{G}) = d$, then by definition, there exists a set $\mathcal{Y}$ with cardinality $d$ such that $\Pi(d, \mathcal{F} \otimes \mathcal{G}) = 2^d$. Therefore, from Eq (36) and (34) we have when $d \geq 2$,

$$2^d = \Pi(d, \mathcal{F} \otimes \mathcal{G}) \leq \Pi(d, \mathcal{F}) \times \Pi(d, \mathcal{G}) \leq \max\{d+1, d^{d_{\mathcal{F}}}\} \cdot \max\{d+1, d^{d_{\mathcal{G}}}\}. \tag{37}$$

For simplicity of notations we denote $d_1 = d_{\mathcal{F}}, d_2 = d_{\mathcal{G}}$ and without loss of generality assume $d_1 \geq d_2$. To complete our proof we need the following auxiliary technical Lemma 3, whose proof can be found in Appendix.

**Lemma 3.** *For any $a \geq 2$ and $m > \frac{1.59a}{\ln 2}(\ln a - \ln \ln 2)$, it holds that $2^m > m^a$.*

Now we are ready to prove our claim. Consider the following situations:

1. if $d_1 \geq 2, d_2 \geq 2$, from Eq (37) we obtain

$$2^d \leq d^{d_1+d_2}.$$

However, from Lemma 3 we know that when $d > \frac{1.59}{\ln 2}(d_1+d_2)(\ln(d_1+d_2)-\ln\ln 2)$, $2^d > d^{d_1+d_2}$ always holds. Hence, in this case we conclude $d \leq \frac{1.59}{\ln 2}(d_1 + d_2)(\ln(d_1 + d_2) - \ln\ln 2) < 2.3(d_1 + d_2)(\ln(d_1 + d_2) + 0.37)$.

2. if $d_1 \geq 2, d_2 = 1$, from Eq (37) we obtain

$$2^d \leq (d+1)d^{d_1} < d^{d_1+2}.$$

From Lemma 3 we know that when $d > \frac{1.59}{\ln 2}(d_1 + 2)(\ln(d_1 + 2) - \ln\ln 2)$, $2^d > d^{d_1+2}$ always holds. Hence, in this case we conclude $d \leq \frac{1.59}{\ln 2}(d_1+2)(\ln(d_1+2)-\ln\ln 2) < 2.3(d_1+2)(\ln(d_1 + 2) + 0.37)$.

3. if $d_1 = d_2 = 1$, from Eq (37) we obtain

$$2^d \leq (d+1)^2.$$

Since $2^m > (m+1)^2$ holds for any $m \geq 6$, we conclude that $d \leq 5 < \frac{1.59}{\ln 2}(2+2)(\ln(2+2) - \ln\ln 2)$.

Combining the three cases, we conclude that

$$PDim(\mathcal{F} \otimes \mathcal{G}) < 2.3(\max\{2, d_1\} + \max\{2, d_2\})(\log(\max\{2, d_1\} + \max\{2, d_2\}) + 0.37)$$
$$< 3(1 + \log d_1 d_2)(d_1 + d_2)$$

$\square$

Now we are ready to prove Theorem 1.

*Proof of Theorem 1.* Classic results from learning theory Pollard (1984) show the following generalization guarantees: Suppose $[0, H]$ is the range of functions in hypothesis class $\mathcal{H}$. For any $\delta \in (0, 1)$, and any distribution $\mathcal{D}$ over $\mathcal{X}$, with probability $1 - \delta$ over the draw of $\mathcal{S} \sim \mathcal{D}^n$, for all functions $h \in \mathcal{H}$, the difference between the average value of $h$ over $\mathcal{S}$ and its expected value gets bounded as follows:

$$\left| \frac{1}{n} \sum_{x \in \mathcal{S}} h(x) - \mathbf{E}_{y \sim \mathcal{D}}[h(y)] \right| = \mathcal{O}\left( H\sqrt{\frac{1}{n}\left(\text{PDim}(\mathcal{H}) + \ln\left(\frac{1}{\delta}\right)\right)} \right)$$

Substituting $\mathcal{H}$ with the class of social distortion mitigation functions $\mathcal{H}_{\mathcal{F}} = \{h(f; \boldsymbol{x}, c) | f \in \mathcal{F}\}$ induced by some moderator function class $\mathcal{F} = f$ gives:

$$\left| \frac{1}{n} \sum_{(\boldsymbol{x}_i, c_i) \in \mathcal{S}} h(f; \boldsymbol{x}_i, c_i) - \mathbb{E}_{(\boldsymbol{x},c) \sim \mathcal{X} \times \mathcal{C}}[h(f; \boldsymbol{x}, c)] \right| = \mathcal{O}\left( H\sqrt{\frac{1}{n}\left(\text{PDim}(\mathcal{H}_{\mathcal{F}}) + \ln\left(\frac{1}{\delta}\right)\right)} \right)$$

Therefore, for a training set $S$ of size $\mathcal{O}\left(\frac{H^2}{\varepsilon^2}[\text{PDim}(\mathcal{H}_{\mathcal{F}}) + \ln(1/\delta)]\right)$, the empirical average social distortion and the average social distortion on the distribution are within an additive factor of $\varepsilon$.

Next, we show for any class $\mathcal{F}$, distribution $\mathcal{D}$ over $\mathcal{X} \times \mathcal{C}$, if a large enough training set $S$ is drawn from $\mathcal{D}$, then with high probability, every $f \in \mathcal{F}$, filters out approximately the same fraction of examples from the training set and the underlying distribution $\mathcal{D}$ had these examples manipulated to their ideal location($\boldsymbol{z}'$). In order to prove this, we use uniform convergence guarantees.

Let $\mathcal{X}' = \{\boldsymbol{x} + \frac{\boldsymbol{e}}{2c} \mid \boldsymbol{x} \in \mathcal{X}, c \in \mathcal{C}\}$ and $\mathcal{Y} = \{0\}$ and consider the joint distribution $\mathcal{D}'$ on $\mathcal{X}' \times \mathcal{Y}$. A hypothesis $f \in \mathcal{F}$ incurs a mistake on an example $(\boldsymbol{x} + \frac{\boldsymbol{e}}{2c}, y)$ if it labels it as positive or equivalently if it filters it out. By uniform convergence guarantees, given a training sample $S'$ of size $\mathcal{O}\left(\frac{1}{\varepsilon^2}[VCDim(\mathcal{F}) + \log(1/\delta)]\right)$, with probability at least $1 - \delta$ for every $f \in \mathcal{F}$, $|\text{err}_{\mathcal{D}'}(f) - \text{err}_{S'}(f)| \leq \varepsilon$. This is equivalent to saying the fraction of points filtered out by $f$ from $\mathcal{D}'$ and $S'$ are within an additive factor of $\varepsilon$.

Here, $\mathcal{F}$ is the class of moderator functions, and $\mathcal{H}_{\mathcal{F}}$ is the class of social distortion mitigation functions induced by $\mathcal{F}$. Now, given a training set $S$ of size $\mathcal{O}(\frac{1}{\varepsilon^2}[H^2(\text{PDim}(\mathcal{H}_{\mathcal{F}}) + \ln(1/\delta)) + VCDim(\mathcal{F})])$, by an application of union bound, for every $f \in \mathcal{F}$, the probability that the average social distortion of $f$ on $S$ and $\mathcal{D}$ differ by more than $\varepsilon$ or the fraction of filtered points differ by more than $\varepsilon$ is at most $2\delta$. This completes the proof. $\qquad\square$

# E  OMITTED PROOFS IN SECTION 5

## E.1  PROOF OF LEMMA 1

*Proof.* Plugin the expression of $\boldsymbol{x}^* = \Delta(\boldsymbol{x}, c; \boldsymbol{e}, f)$ given by Proposition 1 and note that $\Delta(\boldsymbol{x}_i, c_i; \boldsymbol{e}, \perp) = \boldsymbol{x}_i + \frac{\boldsymbol{e}}{2c_i}$, we get a closed form of $DM(f; \mathcal{X})$ as shown below:

$$
\begin{aligned}
DM((\boldsymbol{w}, b); \mathcal{X}) &= \sum_{i=1}^{n} \{D(\perp; (\boldsymbol{x}_i, c_i), \boldsymbol{e}) - D(\boldsymbol{w}, b; (\boldsymbol{x}_i, c_i), \boldsymbol{e})\} \\
&= \sum_{i \in I_0(\boldsymbol{w},b)} \left\| \frac{\boldsymbol{e}}{2c_i} \right\|_2^2 - \sum_{i \in I_1(\boldsymbol{w},b)} \left\| \frac{\boldsymbol{e}}{2c_i} \right\|_2^2 - \sum_{i \in I_2(\boldsymbol{w},b)} \left\| \frac{\boldsymbol{e}}{2c_i} - \frac{\boldsymbol{w}^\top(\boldsymbol{e} + 2c_i\boldsymbol{x}_i)\boldsymbol{w}}{2c_i\boldsymbol{w}^\top\boldsymbol{w}} - \frac{b\boldsymbol{w}}{\boldsymbol{w}^\top\boldsymbol{w}} \right\|_2^2 \\
&= \frac{1}{4}\sum_{i \in I_0} \frac{1}{c_i^2} - \frac{1}{4}\sum_{i \in I_1} \frac{1}{c_i^2} - \sum_{i \in I_2} \left\{ \frac{1}{4c_i^2} + \frac{1}{\boldsymbol{w}^\top\boldsymbol{w}}\left[ (\boldsymbol{w}^\top\boldsymbol{x}_i + b)^2 - \left( \frac{\boldsymbol{w}^\top\boldsymbol{e}}{2c_i} \right)^2 \right] \right\} \\
&= \sum_{i \in I_2} \frac{1}{\boldsymbol{w}^\top\boldsymbol{w}}\left[ -(\boldsymbol{w}^\top\boldsymbol{x}_i + b)^2 + \left( \frac{\boldsymbol{w}^\top\boldsymbol{e}}{2c_i} \right)^2 \right]. 
\end{aligned}
\tag{38}
$$

Here the set $I_0 = \{i \in [n] : \boldsymbol{w}^\top\boldsymbol{x}_i + b \leq 0\}$ contains the indices of all users who are marked as non-problematic and $I_1 = \{i \in [n] : \boldsymbol{w}^\top \cdot \left(\boldsymbol{x}_i + \frac{\boldsymbol{e}}{2c_i}\right) + b \leq 0\} \cap I_0$, $I_2 = \{i \in [n] : \boldsymbol{w}^\top \cdot \left(\boldsymbol{x}_i + \frac{\boldsymbol{e}}{2c_i}\right) + b > 0\} \cap I_0$.

Since a re-scaling of the vector $(\boldsymbol{w}, b)$ does not change the value of the RHS of Eq. (38), we may without loss of generality assume $\|\boldsymbol{w}\|_2 = 1$ and the $DM$ function becomes

$$
DM((\boldsymbol{w}, b); \mathcal{X}) = \sum_{i \in I_2} \left[ -(\boldsymbol{w}^\top\boldsymbol{x}_i + b)^2 + \left( \frac{\boldsymbol{w}^\top\boldsymbol{e}}{2c_i} \right)^2 \right].
\tag{39}
$$

Next, we argue that maximizing Eq. (39) under the constraint $\|\boldsymbol{w}\|_2 = 1$ is equivalent to maximizing it under the constraint $\|\boldsymbol{w}\|_\infty = 1$. This is because, for any solution $(\boldsymbol{w}^*, b^*)$ that yields the optimal value of

Eq. (39) with $\|\boldsymbol{w}^*\|_2 = 1$, re-scaling $(t\boldsymbol{w}^*, tb^*)$ such that $\|t\boldsymbol{w}^*\|_\infty = 1$ would also yield the largest value of the RHS of Eq. (38). And on the other hand, any solution $(\boldsymbol{w}^*, b^*)$ that yields the optimal value of Eq. (39) with $\|\boldsymbol{w}^*\|_\infty = 1$, we can also re-scale it such that $\|\boldsymbol{w}^*\|_2 = 1$. This suggests that we can equivalently consider the objective function given in Eq. (39) and replacing the original constraint $\|\boldsymbol{w}^*\|_2 = 1$ with $\|\boldsymbol{w}^*\|_\infty = 1$.

$\square$

## E.2 PROOF OF THEOREM 2

The $\epsilon$-relaxed version of OP (9) is given by the following:

$$
\begin{aligned}
\text{find} \quad & \arg\min_{\boldsymbol{w},b} \left\{ \sum_{i:\epsilon - \frac{\boldsymbol{w}^\top \boldsymbol{e}}{2c_i} \le \boldsymbol{w}^\top \boldsymbol{x}_i + b \le 0} \left[ (\boldsymbol{w}^\top \boldsymbol{x}_i + b)^2 - \left( \frac{\boldsymbol{w}^\top \boldsymbol{e}}{2c_i} \right)^2 \right] \right\} \\
\text{subject to} \quad & \sum_{i=1}^n \mathbb{I} \left[ \boldsymbol{w}^\top \left( \boldsymbol{x}_i + \frac{\boldsymbol{e}}{2c_i} \right) + b \le 0 \right] \ge n - K, \\
& -1 \le w_j \le 1, 1 \le j \le d.
\end{aligned} \tag{40}
$$

*Proof.* For arbitrary $n$ points $\boldsymbol{y}_1, \cdots, \boldsymbol{y}_n \in \mathbb{R}^d$ and $K < n$, construct an OP (40) instance by letting $\boldsymbol{e} = (0, \cdots, 0, 1)$, $c_1 = \cdots = c_{2n} = \frac{1}{2\epsilon} > 0$, and

$$
\boldsymbol{x}_i = \begin{cases} (y_{i,1}, \cdots, y_{i,d}, 0), & 1 \le i \le n, \\ (0, \cdots, 0, 0), & n+1 \le i \le 2n. \end{cases}
$$

Then solving OP (40) is equivalent to

$$
\begin{aligned}
\text{find} \quad & \arg\max_{\boldsymbol{w},b} \left\{ \sum_{\{1 \le i \le 2n: \epsilon(1 - \boldsymbol{w}^\top \boldsymbol{e}) \le \boldsymbol{w}^\top \boldsymbol{x}_i + b \le 0\}} \left[ -(\boldsymbol{w}^\top \boldsymbol{x}_i + b)^2 + \epsilon^2 \left( \boldsymbol{w}^\top \boldsymbol{e} \right)^2 \right] \right\} \\
\text{subject to} \quad & \sum_{i=1}^{2n} \mathbb{I} \left[ \boldsymbol{w}^\top (\boldsymbol{x}_i) + b \le -\epsilon \boldsymbol{w}^\top \boldsymbol{e} \right] \ge 2n - K, \\
& -1 \le w_j \le 1, \forall 1 \le j \le d+1, \\
& \boldsymbol{w} \ne \boldsymbol{0}.
\end{aligned} \tag{41}
$$

We argue that that the optimal $\boldsymbol{w}^*$ for OP (41) must satisfy $w_{d+1}^* = 1$, because for any $\boldsymbol{w}$ with $w_{d+1} < 1$, we have $\epsilon(1 - \boldsymbol{w}^\top \boldsymbol{e}) > 0$ and thus the set $\{1 \le i \le 2n : \epsilon(1 - \boldsymbol{w}^\top \boldsymbol{e}) \le \boldsymbol{w}^\top \boldsymbol{x}_i + b \le 0\}$ is empty. This means that the objective value of OP (41) is zero. However, any $\boldsymbol{w}, b$ with $w_{d+1} = 1$ such that $\boldsymbol{w}^\top \boldsymbol{x}_i + b = 0$ for some $i$ yields an objective value $\epsilon^2 > 0$. As a result, the optimal $\boldsymbol{w}^*$ that maximize the objective of OP (41) must satisfy $w_{d+1}^* = 1$. Therefore, we can without loss of generality let $\boldsymbol{w}^\top \boldsymbol{e} = 1$ and then solving OP (41) is equivalent to

$$
\begin{aligned}
\text{find} \quad & \arg\max_{\boldsymbol{w},b} \left\{ \sum_{\{1 \le i \le 2n: \boldsymbol{w}^\top \boldsymbol{x}_i + b = 0\}} \left[ \epsilon^2 \right] \right\} \\
\text{subject to} \quad & \sum_{i=1}^{2n} \mathbb{I}[\boldsymbol{w}^\top \boldsymbol{x}_i + b \le -\epsilon] \ge 2n - K, \\
& -1 \le w_j \le 1, \forall 1 \le j \le d+1, \\
& \boldsymbol{w} \ne \boldsymbol{0}.
\end{aligned} \tag{42}
$$

Let $\tilde{\boldsymbol{w}} = (w_1, \cdots, w_d)$ be the first $d$ dimensions of $\boldsymbol{w}$, then solving OP (42) is equivalent to solving the following

$$
\begin{aligned}
\text{find} \quad & \arg\max_{\tilde{\boldsymbol{w}},b} \left\{ n \cdot \mathbb{I}[b = 0] + \sum_{\{1 \le i \le n\}} \mathbb{I} \left[ \tilde{\boldsymbol{w}}^\top \boldsymbol{y}_i + b = 0 \right] \right\} \\
\text{subject to} \quad & \sum_{i=1}^n \mathbb{I}[\tilde{\boldsymbol{w}}^\top \boldsymbol{y}_i + b \le -\epsilon] \ge n - K, \\
& -1 \le \tilde{w}_j \le 1, \forall 1 \le j \le d, \\
& \tilde{\boldsymbol{w}} \ne \boldsymbol{0}.
\end{aligned} \tag{43}
$$

We argue that the optimal solution of OP (43) must satisfy $b^* = 0$. This is because when $b = 0$, any $\tilde{\boldsymbol{w}}$ that satisfies $\tilde{\boldsymbol{w}}^\top \boldsymbol{y}_i + b = 0$ for some $i$ yields an objective value at least $n + 1$. However, if $b \ne 0$, any $\tilde{\boldsymbol{w}}$

in the feasible region would yield an objective value at most $n$. As a result, solving OP (43) is equivalent to solving the following

$$
\begin{aligned}
&\text{find} &&\arg\max_{\boldsymbol{w}} \left\{ \sum_{\{1 \le i \le n\}} \mathbb{I}\left[ \boldsymbol{w}^\top \boldsymbol{y}_i = 0 \right] \right\} \\
&\text{subject to} && \sum_{i=1}^{n} \mathbb{I}[\boldsymbol{w}^\top \boldsymbol{y}_i \le -\epsilon] \ge n - K, \\
& && -1 \le w_j \le 1, \forall 1 \le j \le d, \\
& && \boldsymbol{w} \ne \boldsymbol{0}.
\end{aligned}
\tag{44}
$$

Next, we show that optimizing Equation (44) is an NP-hard problem by showing the following decision problem that we call $\varepsilon$-*maximum feasible linear subsystem (MAX-FLS) with mandatory constraints* is NP-hard. Given $\varepsilon > 0$ and a system of linear equations, with a mandatory set of constraints $A\boldsymbol{x} = b_1$ where $b_1 = [0, 0, \cdots, 0]$ and an optional set of constraints $A\boldsymbol{x} \le b_2$ where $b_2 = [-\varepsilon, \cdots, -\varepsilon]$, and $A$ is of size $d \times n$ and integers $1 \le p \le d$ and $0 \le q \le d$, does there exist a solution $\boldsymbol{x} \in R^n$ satisfying at least $p$ optional constraints while violating at most $q$ mandatory constraints? Our proof is inspired by Amaldi & Kann (1995) that showed MAX-FLS is NP-hard, and we show that even when adding a set of mandatory constraints, it remains NP-hard.

In order to prove NP-hardness, we show a polynomial-time reduction from the known NP-complete *Exact 3-Sets Cover* that is defined as follows. Given a set $S$ with $|S| = 3n$ elements and a collection $C = \{C_1, \cdots, C_m\}$ of subsets $C_j \subseteq S$ with $|C_j| = 3$ for $1 \le j \le m$, does $C$ contain an exact cover, i.e. $C' \subseteq C$ such that each element $s_i$ of $S$ belongs to exactly one element of $C'$?

Let $(S, C)$ be an arbitrary instance of *Exact 3-Sets Cover*. We will construct a particular instance of $\varepsilon$-*MAX-FLS with mandatory constraints* denoted by $(A, b, p, q, \varepsilon)$ such that there exists an *Exact 3-Sets Cover* if and only if the answer to the $\varepsilon$-*MAX-FLS with mandatory constraints* instance is affirmative.

We construct an instance of $\varepsilon$-*MAX-FLS with mandatory constraints* as follows. There exists one variable $x_j$ for each subset $C_j \in C$, $1 \le j \le m$. Equations (45) to (47) are optional and Equations (48) to (50) are mandatory constraints. Equation (45) are coverage constraints to make sure each element in $S$ is covered. Constant $a_{i,j}$ is equal to 1 if $s_i \in C_j$ and is equal to 0 otherwise.

$$
\sum_{j=1}^{|C|} a_{i,j} x_j - x_{m+1} = 0 \qquad\qquad \forall 1 \le i \le 3n \tag{45}
$$

$$
x_j - x_{m+1} = 0 \qquad\qquad \forall 1 \le j \le m \tag{46}
$$

$$
x_j = 0 \qquad\qquad \forall 1 \le j \le m \tag{47}
$$

$$
\sum_{j=1}^{|C|} a_{i,j} x_j - x_{m+1} \le -\varepsilon \qquad\qquad \forall 1 \le i \le 3n \tag{48}
$$

$$
x_j - x_{m+1} \le -\varepsilon \qquad\qquad \forall 1 \le j \le m \tag{49}
$$

$$
x_j \le -\varepsilon \qquad\qquad \forall 1 \le j \le m \tag{50}
$$

We set $p = 3n + m$ and $q = 2n + 2m$. Now, in any solution $\boldsymbol{x}$, we must have $x_{m+1} \ne 0$, since $x_{m+1} = 0$ implies that $x_j = 0$ for all $1 \le j \le m$ in order to have at least $3n + m$ optional constraints satisfied. However, if all $x$ variables are set as 0, then none of the mandatory constraints would be satisfied which is a contradiction.

Now, given any exact cover $C' \subseteq C$ of $(S, C)$, the vector $\boldsymbol{x}$ defined by:

$$
x_j = \begin{cases} -\varepsilon & \text{if } C_j \in C' \text{ or } j = m+1 \\ 0 & \text{otherwise} \end{cases}
$$

satisfies all equations of type Equation (45) and exactly $m$ of Equations (46) and (47). Therefore, $\boldsymbol{x}$ satisfies $3n + m$ optional constraints in total. However, all constraints of type Equation (48) are violated. When $x_j = -\varepsilon$, the mandatory constraint $x_j \leq -\varepsilon$ is satisfied, and $x_j - x_{m+1} \leq -\varepsilon$ is violated. However, when $x_j = 0$, both constraints $x_j - x_{m+1} \leq -\varepsilon, x_j \leq -\varepsilon$ are violated. Since $|C'| = n$, the total number of mandatory constraints violated equals $3n + 2m - n = 2n + 2m$.

Conversely, suppose that we have a solution $\boldsymbol{x}$ that satisfies at least $3n + m$ optional constraints and violates at most $2n + 2m$ mandatory constraints. By construction, since $\boldsymbol{x}$ satisfies $3n + m$ optional constraints, it satisfies all constraints of type Equation (45) and exactly $m$ constraints among Equations (46) and (47) (recall that we are interested in non-trivial solutions, therefore $x_{m+1} \neq 0$). This implies each $x_j$ is either equal to $x_{m+1}$ or $0$. Now, consider the subset $C' \subseteq C$ defined by $C_j \in C'$ if and only if $x_j = x_{m+1}$. This gives an exact cover of $(S, C)$.

Finally, we can conclude that given solution $\boldsymbol{x}$ that satisfies at least $3n + m$ optional constraints and violates at most $2n + 2m$ mandatory constraints, the subset $C' \subseteq C$ defined by $C_j \in C'$ if and only if $x_j = x_{m+1}$ is an exact cover of $(S, C)$.

$\square$

### E.3 Proof of Proposition 4

*Proof.* OP (10) is equivalent to the following problem
$$\text{find} \quad \arg\min_{\boldsymbol{w}, b}\{-DM(\boldsymbol{w}, b) + \lambda P(\boldsymbol{w}, b)\}. \tag{51}$$
And according to the definitions we have
$$(\boldsymbol{w}_1, b_1) = \arg\min_{\boldsymbol{w}, b}\{-DM(\boldsymbol{w}, b) + \lambda_1 P(\boldsymbol{w}, b)\}, \tag{52}$$
$$(\boldsymbol{w}_2, b_2) = \arg\min_{\boldsymbol{w}, b}\{-DM(\boldsymbol{w}, b) + \lambda_2 P(\boldsymbol{w}, b)\}. \tag{53}$$

As a result,
$$\begin{aligned}
-DM(\boldsymbol{w}_2, b_2) + \lambda_2 P(\boldsymbol{w}_2, b_2) \leq & -DM(\boldsymbol{w}_1, b_1) + \lambda_2 P(\boldsymbol{w}_1, b_1) \\
= & -DM(\boldsymbol{w}_1, b_1) + \lambda_1 P(\boldsymbol{w}_1, b_1) + (\lambda_2 - \lambda_1) P(\boldsymbol{w}_1, b_1) \\
\leq & -DM(\boldsymbol{w}_2, b_2) + \lambda_1 P(\boldsymbol{w}_2, b_2) + (\lambda_2 - \lambda_1) P(\boldsymbol{w}_1, b_1)
\end{aligned} \tag{54}$$
Rearranging terms in Eq. (54) we have $(\lambda_2 - \lambda_1)(P(\boldsymbol{w}_2, b_2) - P(\boldsymbol{w}_1, b_1)) \leq 0$, which implies $P(\boldsymbol{w}_1, b_1) \leq P(\boldsymbol{w}_2, b_2)$.

Now we prove $DM(\boldsymbol{w}_1, b_1) \leq DM(\boldsymbol{w}_2, b_2)$. First of all, when $\lambda_2 = 0$, we have $(\boldsymbol{w}_2, b_2) = \arg\max_{\boldsymbol{w}, b} DM(\boldsymbol{w}, b)$ and therefore $DM(\boldsymbol{w}_1, b_1) \leq DM(\boldsymbol{w}_2, b_2)$ must hold. When $\lambda_2 > 0$, it also holds that
$$\begin{aligned}
\frac{\lambda_1}{\lambda_2}\left(-DM(\boldsymbol{w}_2, b_2) + \lambda_2 P(\boldsymbol{w}_2, b_2)\right) \leq & \frac{\lambda_1}{\lambda_2}\left(-DM(\boldsymbol{w}_1, b_1) + \lambda_2 P(\boldsymbol{w}_1, b_1)\right) \\
= & -DM(\boldsymbol{w}_1, b_1) + \lambda_1 P(\boldsymbol{w}_1, b_1) - \frac{\lambda_1 - \lambda_2}{\lambda_2} DM(\boldsymbol{w}_1, b_1) \\
\leq & -DM(\boldsymbol{w}_2, b_2) + \lambda_1 P(\boldsymbol{w}_2, b_2) - \frac{\lambda_1 - \lambda_2}{\lambda_2} DM(\boldsymbol{w}_1, b_1)
\end{aligned} \tag{55}$$

Rearranging terms in Eq. (55) we have $(\lambda_1 - \lambda_2)(DM(\boldsymbol{w}_1, b_1) - DM(\boldsymbol{w}_2, b_2)) \leq 0$, which implies $DM(\boldsymbol{w}_1, b_1) \leq DM(\boldsymbol{w}_2, b_2)$.

$\square$

# F  ADDITIONAL EXPERIMENTS

## F.1  ADDITIONAL DETAILS OF EXPERIMENT

**Synthetic data generation:** We generate synthetic dataset from mixed Gaussian distribution in $\mathbb{R}^d$ to mimic the distribution of $\boldsymbol{x}$. Specifically, we first sample $k$ centers $\mathbf{c}_i$ from $\mathcal{N}(0, I_d)$ and then for each $\mathbf{c}_i$ generate $m = n/k$ samples from $\mathcal{N}(\mathbf{c}_i, \sigma_i^2 I_d)$, where $\sigma_i$ is sampled uniformly at random from $[0.3, 0.5]$. Without loss of generality we set $\boldsymbol{e}$ as the unit vector $(1, 0, \cdots, 0)$, and sample $c_i$ independently from uniform distribution $\mathcal{U}[0.5, 1.5]$. In the experiments we choose $d = 5, n = 500, k = 5$, and additional result under different data scales can be found in Appendix F.

**Optimization solver setup:** we solve OP (11) by setting the $\ell_2$ penalty $P(\boldsymbol{w}, b) = \|g(\boldsymbol{w}, b; U, \boldsymbol{e})\|_2$, i.e., $P_i(\boldsymbol{w}, b) = \mathbb{I}[y_i > 0] \cdot y_i^2$, where $a_i = \frac{\boldsymbol{w}^\top \boldsymbol{e}}{2c_i}, y_i = \boldsymbol{w}^\top \boldsymbol{x}_i + b + a_i$ as defined in the constraints of OP (11). The reason we choose such a form is because the resultant social good loss function $l$ can preserve continuity and first-order differentiable property, paving the way for gradient-based method. Additional details about the optimizer setup can be found in Appendix F.1.

We solve OP (11) by setting the $\ell_2$ penalty $P(\boldsymbol{w}, b) = \|g(\boldsymbol{w}, b; U, \boldsymbol{e})\|_2$, i.e., $P_i(\boldsymbol{w}, b) = \mathbb{I}[y_i > 0] \cdot y_i^2$, where $a_i = \frac{\boldsymbol{w}^\top \boldsymbol{e}}{2c_i}, y_i = \boldsymbol{w}^\top \boldsymbol{x}_i + b + a_i$ as defined in the constraints of OP (11). The reason we choose such a form is because the resultant social good loss function $l$ can preserve continuity and first-order differentiable property, paving the way for gradient-based method. To further differentiate $l$ at $y_i = 0$, we apply spline interpolation at $y_i = 0.1$ to round the corner at $y_i = 0$ while ensuring that $l \to 0$ as $y_i \to -\infty$. The surrogate loss function $l$ for each user $(\boldsymbol{x}_i, c_i)$ after such regularizations compared with the true loss $l$ is illustrated in the leftmost panel of Figure 3, and we can observe that the minimum of $l$ is achieved when $y_i = a_i$, i.e., when the original feature $\boldsymbol{x}$ is on the decision boundary of filter $f$. In addition, for a larger penalty $\lambda$, moving across the boundary (i.e., $y_i$ moving to the right side of $a_i$) would incur a larger and more rapidly increasing loss. The objective is thus the sum of these surrogate losses at all points $(\boldsymbol{x}_i, c_i)$. To account for the boundary constraint $-1 \le w_i \le 1$, we employ the standard projected gradient descent.

The surrogate loss $\tilde{l}(y, a; \epsilon)$ for a single point $(\boldsymbol{x}, c)$ we use in the experiment is given by the following explicit form:

$$\tilde{l}(y, a; \epsilon, \lambda) = \begin{cases} \frac{(1-\epsilon^2)^2 a^3}{2\epsilon y - 4a(1-\epsilon) + 3a(1-\epsilon)^2}, & y < (1-\epsilon)a, \\ y^2 - 2ay, & (1-\epsilon)a \le y \le a, \\ \lambda(y-a)^2 - a^2, & y > a, \end{cases} \tag{56}$$

where $y = \boldsymbol{w}^\top \boldsymbol{x} + b + a, a = \frac{\boldsymbol{w}^\top \boldsymbol{e}}{2c}$, as shown in Figure 4. In our experiments we choose $\epsilon = 0.9$ and use different $\lambda$ ranging from $0.1$ to $100$.

Then we use projected gradient descent (PGD) to solve the following OP 57 with the exact gradient of $\tilde{l}$ w.r.t. $\boldsymbol{w}$ and $b$. The learning rate of PGD is set to $0.1$ and the maximum iteration steps is set to $2000$.

$$\begin{aligned} &\text{find} && \arg\min_{\boldsymbol{w}, b} \left\{ \sum_{1 \le i \le n} \tilde{l}(y_i, a_i) \right\} \\ &\text{subject to} && y_i = \boldsymbol{w}^\top \boldsymbol{x}_i + b + a_i, 1 \le i \le n, \\ & && a_i = \frac{\boldsymbol{w}^\top \boldsymbol{e}}{2c_i}, 1 \le i \le n, \\ & && -1 \le w_i \le 1, 1 \le i \le n. \end{aligned} \tag{57}$$

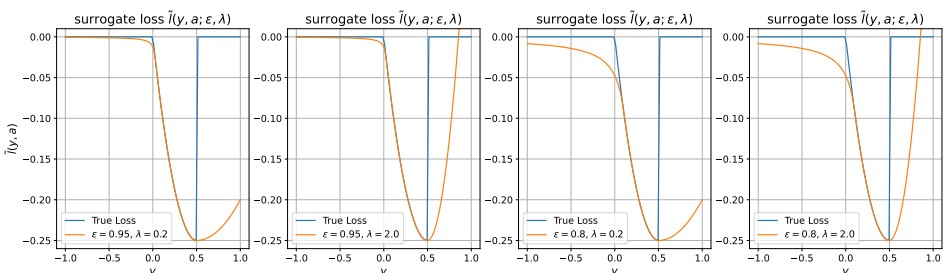

Figure 4: The constructed quasi-convex single point surrogate loss function with different smoothing parameter $\epsilon$ and soft freedom of speech penalty strength $\lambda$. In this illustration we set $a = 0.5$.

## F.2 Additional result under different dimension $d$

We also plot the trade-offs achieved by the computed optimal linear moderators across different dimensions $d$, with the results shown in Figure 5. As the figure illustrates, higher dimensions introduce more noise into the results, but the same underlying insights remain observable.

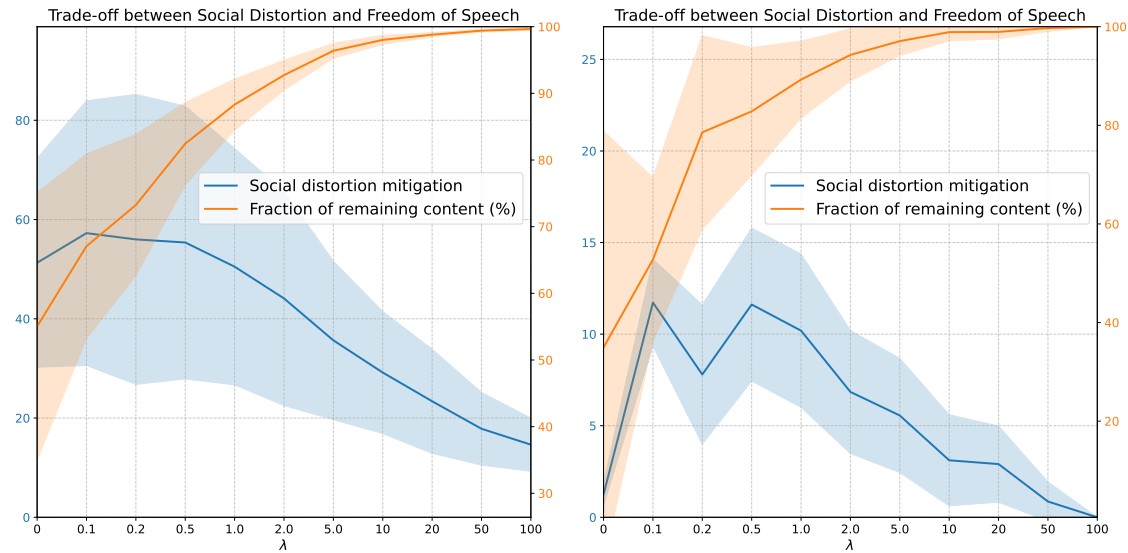

Figure 5: Social distortion mitigation (blue) and the fraction of remaining content on the platform (yellow) incurred by the computed moderator obtained under different $\lambda \in [0.1, 100]$. Left: $d = 2$, Right: $d = 10$. Error bars are $1\sigma$ region based on results from 20 independently generated datasets.