# OpenReview forum: "Strategic Filtering for Content Moderation: Free Speech or Free of Distortion?"
_ICLR.cc/2026/Conference — Submitted to ICLR 2026_

### Official Review · Reviewer_zcN8 · 2025-10-30

**Soundness:** 4
**Presentation:** 4
**Contribution:** 3
**Rating:** 6
**Confidence:** 3

**Summary:**

This paper frames the trade-off between freedom of speech and social distortion challenge on social media as an optimization problem, and aims to tackle the problem of optimizing this balance through optimization techniques. The work reveals that determining the optimal trade-off is NP-hard, and propose effective approximation of the optimal solution.

**Strengths:**

- The paper addresses an important aspect in online social media platform
- The paper provides sound math formulations with theoretical guarantees
- Great writing and easy to follow.

**Weaknesses:**

Thanks for the great efforts to the paper. In general, this is very solid paper. Beyond the discussed pros above, the reviewer raises some concerns for the authors to improve the paper:

1. Contributions.
    - Is the paper the first to propose use strategic ML to model the free sepech and distortion? Appendix A only discusses the practices of content moderation, but does not discuss math frameworks to analyze such phenomenon.
2. Modeling of the optimization problem. The reviewer is curious about explanations about math framework:
    - In the framework, social trend $e$ is a vector, that is independent from the content moderation function. However, as the paper claims, the social trend is somehow motivated by the practices of users who want to “evade platform enforcement”. The reviews believes that the paper would benefit from modeling $e$ as some functions of $f$.
    - In the framework, the users modifies their content by a scalar cost. Besides, each user may have different cost. How is it inspired by the practices on the online platform?
    - In practice, the content moderation function could be a LLM, or LLVM, depending the modality of input. A well-known fact is that these deep-learning-based functions are non-convex. This somehow hurts the implication of this work. Can the authors provide more justifications for this?

**Questions:**

Please respond the concerns as discussed above.

---

### Official Review · Reviewer_Tr5s · 2025-10-31

**Soundness:** 3
**Presentation:** 2
**Contribution:** 1
**Rating:** 2
**Confidence:** 4

**Summary:**

This paper considers the problem of learning a content moderation policy. The goals of the system are to “preserve free speech” and “reduce social distortion”. Technically, this means that the platforms is charged with trying not to induce many shifts from what a user would have posted, except along some vector which they call a “trend”. They prove various results about the computational and statistical complexity of learning policies that solve the stated optimization problem.

**Strengths:**

- The technical results are fairly involved and seem to correctly invoke techniques from statistical learning theory.
- The problem of choosing a content moderation policy is important for the safety and health of online communities.

**Weaknesses:**

- The two goals of the platform are to preserve free speech and to reduce social distortion. These are not well-justified. Platforms are not legally required to preserve free speech, at least in the US, (the first Amendment just prevents the government from restricting speech, not platforms). Some platforms like X seems to market themselves as following free speech principles. Is this the justification for the focus on free speech? Many other platforms have historically not marketed themselves that way. Second, and perhaps more importantly, why would platforms want to reduce the degree to which content producers follow trends? Isn’t this good for the platform, since they might lead to more engagement on the part of users? In most cases, it also doesn’t seem clear that more timely or trend-focused content leads to bad outcomes for users, since they may want to know about or participate in trends themselves. Generally, it seems like the goal of a content moderation policy is to prevent users from being exposed to toxic or illegal content. It doesn’t seem like the trend-reduction goal particularly captures this goal. For example, some content that is toxic, like hateful speech, may not have any trend component to it. At a high level, the model seems to assume that the harms of moderation are relative, in the sense that a change in the direction of some “trend” is bad for the platform. But it seems much more reasonable to model content toxicity as absolute, in the sense that some region in the speech embedding space is toxic and some of it is nontoxic (or other ways of modeling toxicity based on position, rather than distortion from some original intention). Generally, I would need much more clarity on the justification for these goals and modeling choices before considering a positive review.
- The technical results are not particularly informative about the specific context of content moderation. Instead, they are mostly about standard statistical learning theoretic properties of the optimization problem, and seem to mainly follow standard techniques. Proposition 1 is a fact about projections onto convex sets. Proposition 2 is a fact about how moderation policies induce users to change their behavior. Theorem 1 is a sample complexity bound for the moderator learning task, which assumes access to unmanipulated content (when is this realistic? The suggestion for “removing part of the content that violates the guidelines” doesn’t make sense to me.) Second 5 considers linear policies, but these are also not well motivated.
- There is a rich theory literature on content moderation and policy choices, which this paper does not discuss or compare itself to. See, e.g., the following. These are just examples. There are many more papers on this topic.

Mohamed Mostagir and James Siderius. 2023. When Should Platforms Break Echo Chambers? 2023.

Mostagir, Mohamed, and James Siderius. "Naive and Bayesian Learning with Misinformation Policies."

Cynthia Dwork, Chris Hays, Jon Kleinberg, and Manish Raghavan. 2024. Content Moderation and the Formation of Online Communities: A Theoretical Framework.

**Questions:**

What role is the social trend playing? Why is this an element of the model, rather than considering moderation policies about the absolute location of speech, rather than their "social distortion"?

---

### Official Review · Reviewer_L6o5 · 2025-11-04

**Soundness:** 3
**Presentation:** 2
**Contribution:** 3
**Rating:** 4
**Confidence:** 3

**Summary:**

In this work, the authors address the challenge of reducing social distortion while simultaneously preserving freedom of speech on social media platforms. They propose a novel model in which social media users aim to align with current trends while maintaining their original content and adhering to moderation boundaries. From a social media perspective, the authors formulate this as a mechanism design problem, where the goal is to limit social distortion while preserving freedom of speech. They show that optimizing between these two objectives is NP-hard and propose methods for finding approximate solutions.

**Strengths:**

Strengths:
1. The paper addresses an important and timely question of content moderation while aiming to preserve freedom of speech.
2. The proposed model is novel, offering a new perspective on how to mitigate social distortion without over-restricting user expression.
3. The paper presents strong and non-trivial technical results, particularly the generalization result (Theorem 1) and the NP-hardness proof (Theorem 2), which makes the theoretical contributions of this paper strong.

**Weaknesses:**

Weaknesses:
While this paper provides strong theoretical results, the motivation and contextual framing related to content moderation raise some concerns:
1. Lack of discussion of related work:
    * The paper does not reference existing research on content moderation and game theory (even more specifically stackelberg games), despite there being relevant prior work. For example:
        * Optimal Signaling of Content Accuracy: Engagement vs. Misinformation – Ozan Candogan
        * A Persuasive Approach to Combating Misinformation – Safwan Hossain et al.
    * It seems that the whole section comparing the proposed approach with these prior works is missing. Even if the setting differs, referencing, commenting and contrasting relevant literature should be part of this paper.
2. Lack of a coherent story connecting theory to content moderation:
           While the paper is motivated by reducing distortion on social media, and hence the whole proposed model is framed around this motivation, the connection between the theoretical results and their practical relevance is not clearly maintained throughout the paper. As a result, the paper reads more like a purely theoretical work. This seems to be a major weakness given the stated motivation and the immediate social significance of the topic. Since a completely new setting is proposed, it would have been valuable to include more interpretation and commentary on the results that describe their relevance beyond abstract understanding. On the other hand, this would strongly justify the newly proposed setting and the results obtained.

In short, while the theoretical contributions of this paper are strong, this work would gain another dimension in terms of contributions if there were a clearer connection to real-world applications. Furthermore, referencing past work should be included.

**Questions:**

Is there a specific reason why the paper does not reference or discuss prior work on content moderation and game theory?

---

### Official Review · Reviewer_yDzQ · 2025-11-05

**Soundness:** 3
**Presentation:** 1
**Contribution:** 2
**Rating:** 2
**Confidence:** 4

**Summary:**

This paper proposes a method that automatically moderates user-generated content on social media to balance freedom of speech and what appears to be conceived as a reduction of harmfulness. A problem is defined that is NP-hard and a practical approximation is given using statistical learning theory. The idea is that users strive to express themselves within the confines of the guidelines, and yet the platform would still be interested to restrict their expression nevertheless.

**Strengths:**

S1. Interesting mathematical problem formulation.
S2. Solid application of statistical learning theory within the confines of the formulated problem.
S3. Experimentation with a synthetic data set.

**Weaknesses:**

W1. Fundamentally false description of the tradeoff.

W2. Questionable ethical premises.

W3. Lack of real-world examples, motivation, and experimentation.

W4. The fundamental concepts of "trend" and "distortion" are ill-defined: it is claimed the trending direction e "may deviate from users’ true expressive intent"; it is also claimed that if a user x follows a social trend e and their content gets filtered out, then their freedom of speech is violated. These two statements cannot both be true. A user's true expressive intent and a user's freedom of speech are aligned goals. If a trend were deviating from a user's true expressive intent, then that trend would violate their freedom of speech.

**Questions:**

Why do you think there is a tradeoff between freedom and non-distortion?

**Details Of Ethics Concerns:**

The paper is founded on questionable ethical premises and false assumptions.
In particular, starting with the abstract, it is proposed that a tradeoff exists between:
(1) ensuring freedom of speech, and
(2) reducing social distortion.
However, no such tradeoff exists: the more one ensures freedom of speech, the more one reduces social distortion as well, since freedom and veracity are inherently compatible. The foundational premise of the paper appears thus to be false.

It seems, however, that authors may have meant something else. The goal may be to balance the freedom of speech, on the one hand, and the distortion presumably necessary to prevent harmful speech. If that were the goal, then it should have been stated so. Even assuming that this is the goal, nevertheless, still leaves the paper on questionable ethical grounds. There is something fundamentally odd about the very idea that freedom and truth are expressible as mathematical objects subject to vector optimization. The paper never enters into a discussion on why such premises would hold and what they would amount to in the real world.

---

### Meta-Review · Area_Chair_W4gd · 2026-01-04

**Summary:**

Reviewers raised serious concerns about the paper’s foundational assumptions, modeling choices, and relevance. Multiple reviewers (Reviewers yDzQ and Tr5s) noted that the core concepts of trend, distortion, and freedom of speech are ill-defined and not inconsistent, with the central premise of a tradeoff between freedom of speech and distortion being ethically and conceptually questionable (potentially even false). Reviewers also highlighted a weak connection between the theoretical framework and real-world content moderation, making the work read as largely abstract to me with unclear practical implications and potential for real-world implementation (Reviewers L6o5 and Tr5s). Technically, I agree with Reviewer Tr5s that the results are fairly standard invocations of statistical learning theory with regards to the formulated optimization problem and are not particularly informative for the content moderation problem. In the same vein, there were several open questions about the modeling of the optimization problem that went unanswered (Reviewer zcN8). Finally, the paper lacks adequate engagement with a substantial body of related work on content moderation, game theory, and ML in social media safety, further limiting its contribution (Reviewers L6o5 and Tr5s). Moreover, the authors did not engage with the reviewers during the rebuttal period in any capacity.

**Reviewer Concerns:**

None of the concerns were addressed, as the authors did not engage in the rebuttal.

**Reviewer Scores:**

As the authors did not engage with reviewers, I do not believe reviewers would have changed their scores, as their concerns/issues are well justified.

---

### Decision · Program_Chairs · 2026-01-26

Reject